

# Seasonal and interannual variability of water column properties along the Rottnest continental shelf, south-west Australia

Miaoju Chen[1], Charitha B. Pattiaratchi[1], Anas Ghadouani[2] and Christine Hanson[1]

[1]Oceans Graduate School & the UWA Oceans Institute, the University of Western Australia, Perth, 6009, Australia
[2]School of Engineering, the University of Western Australia, Perth, 6009, Australia

*Correspondence to:* Miaoju Chen (miaoju.chen@research.uwa.edu.au)



**Abstract**

A multiyear, ocean glider dataset, obtained along a representative cross-shelf transect along the Rottnest continental shelf, south-west Australia, was used to characterise the seasonal and inter-annual variability of water column properties (temperature, salinity, and chlorophyll fluorescence distribution). All three variables showed distinct seasonal and inter-annual variations. Local and basin-scale ocean–atmosphere processes also affected the spatial distributions of the water column properties. The controlling influences for the variability were derived from (a) at the local scale, the Leeuwin Current and dense shelf water cascades (DSWC); and, (2) at the basin scale, the El Niño Southern Oscillation (ENSO). In spring and summer, shallow waters were well mixed due to strong wind mixing and the deeper waters (> 50 m) were vertically stratified in temperature that contributed to the formation of a subsurface chlorophyll maximum (SCM). With the onset of storms in late autumn, the water column was well mixed with the SCM absent. On the inner shelf, chlorophyll fluorescence concentrations were highest in autumn and winter; DSWCs were also the main physical feature during autumn and winter. Chlorophyll fluorescence concentration was higher closer to the sea bed than at the surface in spring, summer, and autumn. The seasonal patterns coincided with changes in the wind field (weaker winds in autumn) and air–sea fluxes (winter cooling and summer evaporation). Inter-annual variation was associated with ENSO events. Lower temperatures, higher salinity, and higher chlorophyll fluorescence (> 1 mgm$^{-3}$) were associated with the El Niño event in 2010. During the strong La Niña event in 2011, temperatures increased (a 'marine heat wave'), and salinity and chlorophyll fluorescence decreased (< 1 mgm$^{-3}$). These changes were mainly associated with changed to the strength of the Leeuwin current. Over subsequent years, the temperatures gradually decreased, the salinity increased, and the chlorophyll fluorescence continued to decrease (< 0.25 mgm$^{-3}$). These changes were mainly associated with an increase in the strength of the Leeuwin current that transported warmer, lower salinity, low nutrient water into the region. In the autumn of 2014, the chlorophyll fluorescence increased (> 1 mgm$^{-3}$). It is concluded that the observed seasonal and inter-annual variability in chlorophyll fluorescence concentrations were related to the changes in physical forcing (wind forcing, Leeuwin Current and air-sea fluxes).



## 1. Introduction

Almost all life forms rely on primary production, directly or indirectly, to survive, and phytoplankton in the ocean perform most of the primary production (Field et al., 1998). The phytoplankton pigment, chlorophyll, is an important biological indicator of phytoplankton biomass in the water column. Environmental variables, such as light availability (Sverdrup, 1985; Huisman and Weissing, 1994), water temperature (Eppley, 1992; Hambright et al., 1994; Paerl and Huisman, 2008), and salinity (Karsten et al., 1995), affect phytoplankton biomass variability. Seasonal cycles of phytoplankton concentrations signify the annual growth activity in pelagic systems (Cebrián and Valiela, 1999; Winder and Cloern, 2010). The most common cycle is the spring bloom—an increase in phytoplankton concentrations in response to seasonal changes in temperature and solar radiation—which is usually present for a few weeks to months (Cushing, 1959; Sommer et al., 1986). Often a secondary peak develops in late summer or autumn (Longhurst, 1995). These seasonal phytoplankton patterns have large inter-annual variability across different geographic regions (Cebrián and Valiela, 1999; Cloern and Jassby, 2008; Garcia-Soto and Pingree, 2009). Satellite and field-based measurements have shown that in the oligotrophic waters off south-west Australia, the seasonal chlorophyll cycle (a proxy for phytoplankton biomass) is different from that in other regions, with a clear peak in chlorophyll concentrations in late autumn or early winter and minimal levels in spring and summer (Koslow et al., 2008; Thompson et al., 2011; Lourey et al., 2012). Pearce et al. (2000) found higher chlorophyll concentrations on the continental shelf than further offshore. In this paper, we present an extensive, multiyear, ocean glider dataset, obtained along a representative cross-shelf transect along the Rottnest continental shelf, south-west Australia, to explore the seasonal and inter-annual variability of water column properties (temperature, salinity, and chlorophyll fluorescence distribution). Although ocean gliders have been sampling the oceans for more than a decade, sustained observations addressing the variability at the seasonal and inter-annual scales from continental shelf regions are almost non-existent and this study addresses this shortcoming.

The Rottnest continental shelf (RCS) has several distinct bathymetric features (Figure 1a): (1) a shallow inshore region (depths < 10 m), which can be defined as a 'leaky' coastal lagoon with a line of discontinuous submerged limestone reefs; (2) an upper continental shelf terrace, with a gradual slope and a mean depth of ~40 m, located from ~10–40 km from the coast; (3) an initial shelf break at the 50 m isobath; (3) a lower continental shelf between the 50 and 100-m isobaths, where the depth increases sharply; and, (4) the main shelf break at the 200 m isobath.

The major current systems in the region are the Leeuwin and Capes currents (Woo and Pattiaratchi, 2008; Wijeratne et al., 2018). The Leeuwin current (LC) is a warm, lower salinity, poleward-flowing, eastern boundary current, which mainly flows along the 200 m isobath (Ridgway and Condie, 2004; Pattiaratchi and Woo, 2009). In this oligotrophic environment lower chlorophyll and nutrient concentrations (Hanson et al., 2005a; Twomey et al., 2007) and lower primary productivity (Hanson et al., 2005b; Koslow et al., 2008) characterise the LC. The LC, which is strongest in autumn and winter, transport ~5–6 Sv of water during austral winter and ~2 Sv the austral summer poleward (Feng et al., 2003; Wijeratne et al., 2018). El Niño and La Niña cycles influence the Leeuwin current: the current is weaker during El Niño events and stronger during La Niña events (Pattiaratchi and Buchan, 1991; Wijeratne et al., 2018). Of interest to this study, the region experienced a





marine heat wave in February and March 2011, which was associated with the warming related to the La Niña event defined as Ningaloo Niña (Feng et al., 2013). This event increased the Leeuwin current's volume transport in February—an unusual event at this time of the year—and resulted in unprecedented warm sea surface temperature anomalies (~5 ℃ higher than normal) off Australia's west coast (Feng et al., 2013).

The Capes current, which is dominant in summer, is a wind-driven, inner shelf current, generally formed in water depths < 50 m (Gersbach et al., 1999). It transports cooler, upwelling-derived water northward past Rottnest Island (Figure 1) between October and March (Pearce and Pattiaratchi, 1999; Gersbach et al., 1999).

Continental shelf processes along the Rottnest continental shelf are mainly wind driven, given the low diurnal tidal range (<0.6 m) (Pattiaratchi and Eliot, 2008). Three wind systems dominate this region: sea breezes; storms (wind speeds >17 m/s), and calm periods (wind speeds <5 m/s) (Verspecht and Pattiaratchi, 2010). Local sea breezes, superimposed on synoptic southerly winds (with speeds often >15 m/s), are prevalent in austral summer and spring (September-February) (Pattiaratchi et al., 1997). Storm systems are most frequent during winter (June-August), and are associated with the passage of frontal systems with the region subject to peak wind speeds to 30 m/s. These storm winds are generally north-westerly in winter and southerly in summer (Verspecht and Pattiaratchi, 2010). In the study region winter storms have a typical pattern with strong north/north-easterly winds blowing for 12 to 52 hours, followed by a period of similar duration when winds turn south/south-westerly, with no prevailing direction dominating for the duration of the storm. Summer storms have southerly winds over a period of 3-4 days that are enhanced by the sea breeze system in the afternoon. Calm wind conditions are mainly observed during autumn and winter (March-August; between winter storm fronts) and are characterized by low wind speeds (<5 m/s).

Another major feature of the dynamics is the presence of dense shelf water cascade (DSWC) on the continental shelf (Pattiaratchi et al., 2011). Western Australia experiences high evaporation rates resulting in higher salinity (density) water in the majority of shallow coastal waters. This dense water is transported across the continental shelf close to the seabed due to the density difference between the nearshore and offshore water (Pattiaratchi et al., 2011). It is major is a mechanism for the export of water containing suspended material and chlorophyll away from the coastal zone. Analysis of ocean glider measurements by Pattiaratchi et al. (2011) indicated that DSWC is a regular occurrence along the RCS particularly during autumn and winter months. In autumn the dense water formation is mainly through changes in salinity resulting from evaporation whilst in winter, temperature was dominant through surface cooling. In summer, although there is a cross-shelf density (due to salinity) gradient, DSWC is not present due to wind induced vertical mixing.

Changes in phytoplankton biomass at seasonal and inter-annual timescales are important components of the total variability associated with biological and biogeochemical ocean processes (Ghisolfi et al., 2015). The circulation along the Western Australian coast has been studied through observations and the use of ocean models (Gersbach et al., 1999; Feng et al., 2003; Woo and Pattiaratchi, 2008; Wijeratne et al., 2018); however, methods to study the biological processes in the water column have been limited to the use of satellite ocean colour data (Moore et al., 2007; Lourey et al., 2012) and shipborne observations (Hanson et al., 2005a; Pearce et



al., 2006; Koslow et al., 2008). Satellite imagery is limited to processes at the sea surface as sensors are unable to images subsurface waters due to limitations in light penetration. Information on the role of physical forcing and the biological responses in the water column has been limited because of the absence of a comprehensive observational dataset. Most of the available oceanographic and biological data are restricted in time and space and are thus unsuitable to be used to study patterns across the RCS at different time and space scales.

Eastern boundaries of ocean basins are typically associated with upwelling of higher nutrient water into the euphotic zone leading to high primary productivity on the continental shelf and rich coastal fisheries (Codospoti, 1983). Oceanographic conditions off RCS are dominated by the LC that suppress upwelling and transports nutrient poor water along the continental shelf and has a negative impact on primary productivity (Koslow et al., 2008; Twomey et al., 2007). The absence of upwelling and major river systems means that the region is low in nutrients. For example, Twomey et al. (2007) reported dissolved nitrate concentrations on continental shelf and in the surface 50m further offshore were typically below detection levels (<0.016 μM). Nitrate concentrations increased rapidly beyond 150 m depth to concentrations of around 30 μM. Thus the RCS is a highly nutrient poor environment with nutrient supply limited to that through recycling during storms and offshore supply through eddy activity (Koslow et al., 2008).

The use of ocean gliders as an observational tool has several advantages over traditional, ship-based surveys: ocean gliders have high sampling frequencies and long sampling durations (Pattiaratchi et al., 2017); the high temporal and spatial resolution data obtained with gliders may provide a better understanding of the links between the physical (meteorological and oceanographic) forcing and the phytoplankton response; all the relevant data are collected simultaneously and are not weather limited. We used a multiyear (2009–2015) ocean glider dataset (50 individual missions) along a repeated transect to examine the variability in the physical parameters and chlorophyll fluorescence concentration (a measure of phytoplankton biomass) distribution over seasonal and inter-annual timescales. The seven-year timescale included two El Niño events (2010 and 2014) and a strong, extended La Niña event (2011–2013). The aims of this paper, through the analysis of the long-term ocean glider dataset, were to: (1) examine the seasonal and inter-annual variability in chlorophyll fluorescence along the Rottnest continental shelf; (2) relate the seasonal chlorophyll fluorescence variability to changes in temperature and salinity distribution and local wind forcing; and, (3) determine the influence of the ENSO cycles on chlorophyll fluorescence. This is the first long term study of seasonal processes in the continental shelf waters along the RCS. Understanding the seasonal and inter-annual variability of coastal ocean properties including how changes in the physical parameters (temperature and salinity) influence chlorophyll distribution across the continental shelf. This enables the identification of the main mechanisms that drive the variations in phytoplankton, as represented by chlorophyll fluorescence along the RCS.

This paper is organised as follows: Section 2 describes the methods. The results of the seasonal winds and the monthly, seasonal, and inter-annual variations in the chlorophyll and physical properties are described in Section 3. In Section 4, we discuss the possible causes of the observed variability. A general conclusion is then given in Section 5.



## 2. Methods

Water column data were obtained from repeated surveys undertaken using Teledyne Webb Research Slocum electric gliders (http://www.webbresearch.com/) along the Two Rocks transect off Rottnest continental shelf, south-west Australia (Figure 1). The Slocum ocean glider is 1.8 m long, 0.213 m in diameter and weighs 52 kg. It is designed to operate in coastal waters of up to 200 m deep where high manoeuvrability is required under relatively strong background currents. Ocean gliders are autonomous underwater vehicles that propel themselves through the water by changing its buoyancy relative to the surrounding water (Rudnick, 2016). By alternately reducing and expanding their volume, ocean gliders can descend and ascend through the ocean water column using minimal energy. In contrast to other similar automated ocean sampling equipment (e.g. Argo floats; http://www.argo.ucsd.edu/), ocean gliders have wings, a rudder and a movable internal battery pack allowing them to move horizontally in a selected direction in a saw tooth pattern. This allows for the horizontal position to be controlled and to sample particular regions of the ocean. The gliders are remotely controlled via the Iridium satellite system and navigate through waypoints, fixing their position via the Global Positioning System (GPS). Each time the glider surfaces, the data and new waypoints can be relayed via satellite to and from the glider. The autonomous nature of the ocean gliders means that they are able to collect data continuously irrespective of the weather conditions.

The data set were collected over a seven year (2009–2015) period. Personnel at the Australian National Facility for Ocean Gliders at the University of Western Australia operate the gliders (Pattiaratchi et al., 2017). All the ocean glider data are available through the Australian Ocean Data Network (https://portal.aodn.org.au). More than 200 cross shelf transects from ~50 ocean glider missions were analysed, with ~28 million individual scans obtained for each variable (temperature, salinity and chlorophyll).

Each cross shelf ocean glider transect took two to three days to complete (Pattiaratchi et al., 2011), with the gliders travelling at a mean speed of 25 km/day. The glider transects extended from ~20 m depth contour to deeper waters (the gliders have a maximum depth range of 200 m) and collected data from the surface to ~2 m above the sea bed. The gliders were equipped with a Sea-Bird Scientific pumped CTD (conductivity–temperature–depth) sensor, a WETLabs BBFL2SLO 3 parameter optical sensor (which measured chlorophyll fluorescence, coloured dissolved organic matter, and backscatter at 660 nm), and an Aanderaa oxygen optode. All the sensors sampled at 4 Hz (which yielded measurements about every 7 cm in the vertical). The actual vehicle trajectory was transposed onto the Two Rocks transect as a straight line (Pattiaratchi et al., 2011).

IMOS data streams are provided in NetCDF-4 format with ocean glider data files containing meta-data and scientific data for each glider mission. Subsequent to the ocean glider recovery, all the data collected by the glider are subject to QA/QC procedures that include a series of automated and manual tests (Woo, 2017). To maintain data integrity all of the sensors (CTD and optical sensors) are returned to the manufacturers for calibration after a period 365 days in the water. The Sea-Bird Scientific SBE 41CP pumped CTD sensor on the Slocum gliders is the same as those mounted on Argo floats and achieve temperature and salinity accuracies of ±0.002 °Celsius and ±0.01 salinity units, respectively. As part of the QA/QC procedures applied to the WETLabs BBFL2SLO 3 Eco Puck sensor on the Slocum gliders are subject to a series tests to track their performance, to





identify faults and to quantify drift during missions due to biofouling and/or damage. These tests were undertaken in the laboratory prior to shipping of the glider, immediately prior to deployment on the vessel, and then immediately on recovery before after cleaning of the sensor face. The tests are carried out by attaching a solid standard in a holder a set distance from the sensor face and collecting engineering counts from the

fluorescence, CDOM and backscatter signals over a 5 minute period. The solid standards used for fluorescence and CDOM counts, Plexiglas Satinice® plum 4H01 DC (polymethylmethacrylate, Evonik Industries), was identified by Earp et al. (2011) in a review of fluorescent standards for calibration of optical instruments as being the optimum. The ocean glider deployed started in 2008 and performance of ECO Puck sensors has been documented over this period. This included comparing consecutive scale factors following factory calibrations.

Our records demonstrated the inherent stability of these sensors in their fluorescent and backscatter measurements, with the difference between fluorescence scale factors between calibrations over 8 years of service typically < 6%. To measure the reliability of the instruments between factory calibrations, the fluorescent response of the instruments to a fluorescein concentration curve have been tracked between factory calibrations ensure ongoing reliability. Field data, obtained from the north-west shelf of Australia, for direct

comparisons between fluorescence and Chlorophyll a (extracted using acidification techniques) indicated a correlation squared ($r^2$) value of 0.75 (Thomson et al., 2015).

Wind speed and direction, recorded at 30 min intervals, were obtained from the Bureau of Meteorology weather station at Rottnest Island, and located ~ 40 km south of Two Rocks transect (Figure 1a).

The focus of this paper is on the seasonal and inter-annual variability in the temperature, salinity, and chlorophyll fluorescence concentrations across the Two Rocks transect. It was assumed that processes along this transect were representative of the cross-shelf variability across the Rottnest continental shelf. When we refer to the 'chlorophyll concentration' (units mg/m³), we are referring to the chlorophyll fluorescence as

recorded by the BBFL2SLO optical sensor. Salinity is expressed as a dimensionless quantity.

When examining the seasonal changes it was found that the changes in the mean values obscured the seasonal variability of each parameter (temperature, salinity, and chlorophyll). Hence, in addition to presenting the measured values we also calculated anomalies to remove the influence of the seasonal variability. The procedure

for each parameter (~28 million individual points) was as follows: (1) data were interpolated onto a common grid across the cross shelf transect; (2) transects were then sorted according to season: spring (September-November), summer (December-February), autumn (March-May) and winter (June-August); (3) the mean value across the whole transect (i.e. through water depth and distance) for each season was calculated; and, (4) the anomaly at each grid point was calculated by subtracting the seasonal mean from values at each point.



## 3.  Results

### 3.1  Seasonal winds

The mean winds for each season from March 2010 to March 2014 showed southerly winds were prevalent in summer, autumn, and spring, followed by south/south-easterly winds (Figure 2). Summer storms, which usually
lasted 36 hours, caused strong, southerly winds (> 25 m/s). The sea breeze usually contributed to the southerly winds, which reinforced the prevailing southerly winds found in the seasonal rose plots (Figure 2). In autumn, the wind speeds decreased (< 13 m/s), whereas in winter, the winds had no prevailing direction. (This is typical of winter storms, which are associated with rapid changes in the wind direction; Verspecht and Pattiaratchi, 2010). In spring, the winds were southerly, with mean wind speeds of 15 m/s.  The southerly winds in the study
region are upwelling favourable.

The time series of the daily mean wind speeds and directions in 2010 revealed the changes that occurred in the wind regime: from November to May (the summer regime), the winds were generally southerly, and the mean wind speeds were ~7.5 m/s in November and ~10 m/s in January and February (Figure 3). The wind speeds
decreased between March and mid-May (the autumn regime), with few changes in the wind direction. Between mid-May and October (the winter regime), winter storms caused large fluctuations in the wind speed and direction.

### 3.2  Seasonal temperature, salinity, and chlorophyll distribution

Typical cross-shore distributions of the seasonal variation in temperature, salinity, and chlorophyll along the
Two Rocks transect during spring, summer, autumn, and winter are shown in Figure 4.

In spring (21–23 October 2013), the temperature and salinity in the upper 80 m were vertically mixed across the whole shelf. The temperature and salinity characteristics changed at the shelf break. On the upper continental shelf (< 40 m depth), the water was cooler and less saline than at lower depths (Figure 4a). High chlorophyll
concentrations (up to 1.2 mg/m³) were found on the inner shelf and at the 50 m shelf break (Figure 4a), which corresponded to temperature and salinity gradients (i.e. frontal system) in the same region. A thin layer (< 10 m) of subsurface chlorophyll maximum layer (SCM) (up to 1 mg/m³) extended from the shelf break to offshore, and coincided with the halocline (and pycnocline) at about 80m  depth.

In summer (28 February–3 March 2014), a plume of warm Leeuwin current water (~23.5 °C) was located in the
top ~30 m depth between 60 and 70 km offshore (Figure 4b). This plume cooled (to 23 °C) and thinned (in the top ~5 m depth) as it moved inshore. Water on the upper continental shelf was cooler than that offshore. The salinity on the upper continental shelf (~35.7) was slightly higher than offshore (35.5); the Capes current most likely caused the cooler and saltier inshore waters. The cooler and saltier water on the upper continental shelf
revealed that higher density water was present on the shelf and a small, dense shelf water cascade (DSWC) was present inshore of the shelf break. A subsurface chlorophyll maximum (up to 1.2 mg/m³, between 50 and 110 m depths (at the pycnocline), was located from the shelf break to offshore.

In autumn (18–21 May 2009), the nearshore waters were saltier and cooler (21 °C) than the offshore waters





(22.5 °C) (Figure 4c). The offshore waters were well mixed except in the bottom 20m. A plume of salty (35.7) and cooler (21 °C) water, which extended to ~60 km across the entire continental shelf and depths > 180 m, was observed inshore and indicated the presence of a DSWC. The maximum chlorophyll concentration (1.3 mg/m³) was located on the upper continental shelf in the shallow pycnocline. The chlorophyll levels were generally

higher in the DSWC than they were in the surface waters on the upper continental shelf. In the offshore waters, higher chlorophyll water was uniformly distributed in the surface mixed layer to depths of 60 m close to the shelf break and depths > 120 m farther offshore (Figure 4c).

In winter (9–11 August 2012), the temperature increased from 18 °C inshore to 20.7 °C offshore, and the water

column was generally vertically mixed (Figure 4d), except between 10 and 20 km on the upper continental shelf. The salinity was uniformly distributed inshore and in most offshore regions. Maximum chlorophyll (> 1 mg/m³) concentrations were found along the inner shelf between 10 and 20 km and corresponded to the region of vertical and horizontal temperature and salinity gradients.

The ocean glider data indicated vertical and horizontal stratification across the shelf and the temperature and salinity distribution across the shelf changed seasonally. The temperature and salinity characteristics on the upper continental shelf were often different from those farther offshore. High chlorophyll concentrations were found in regions with strong temperature and salinity gradients and thus density. These maximum values occurred in the vertical (e.g. the SCM in summer and autumn) and the horizontal (e.g. at the shelf break in

spring and winter).

### 3.3 Monthly mean water masses and chlorophyll concentrations

The monthly mean temperature and salinity were calculated year round, except during July, September, November, and December because only a single ocean glider transect was available in each of these months. The temperature and salinity structure showed that from January to March (summer to early autumn), the

inshore waters (< 40-m depth) were warmer (~21.1–23.0 °C) and the salinity decreased (35.81 to 35.77) (Figure 5a). From March to August (autumn to winter), the temperature (23.0 to 19.0 °C) and salinity both decreased (35.77 to 35.22). From August to January (winter to early summer), both temperature (18.9 to 21.1 °C) and salinity increased (35.22 to 35.8). Offshore water (> 40 m depth) showed a similar seasonal pattern to that of the inshore waters (Figure 5b). Except that from January to March, the salinity decrease in offshore waters (from

35.71 to 35.54) was larger compared to inshore waters and from August to January, the offshore water temperature dropped slightly before increasing to 21.0 °C.

Spatially averaged chlorophyll concentrations for the inshore waters revealed significant seasonal variability. Higher chlorophyll concentration values were reached between March and August (autumn to winter), with a

maximum of 1.12 mg/m³ reached in May and decreased to a minimum of 0.36 mg/m³ in February. The chlorophyll concentration values for offshore waters were less variable with highest values in May (maximum of 0.85 mg/m³) and lowest in February (minimum of 0.43 mg/m³). Higher chlorophyll concentrations corresponded with warmer and less saline water for both the inshore and offshore waters.





### 3.4    Seasonal distribution

Over the seven-year study period (January 2009–March 2015), the vertical structure of the seasonal mean data for the Rottnest continental shelf revealed variability in the temperature, salinity, and chlorophyll concentrations. Anomalies are defined as departures from the seasonal average, with positive (negative) values higher (lower) than the seasonal average. The anomalies allowed us to examine the relative changes in water properties across the whole transect for each season.

*Temperature anomaly*

Seasonal temperature anomaly in the continental shelf waters differed from those further offshore, seaward of the shelf break (Figure 6). During spring, the temperature anomaly indicated that water to be vertically well mixed on the continental shelf (Figure 6a) with warmer water offshore. In summer, the warmer surface water extended across the entire continental shelf (Figure 6b). Water along the middle of the shelf (5–20 km from the coast) was slightly cooler, most likely due to the influence of the Capes Current. In autumn and winter, the upper shelf waters were cooler than the offshore waters. The temperature anomalies were mostly negative, with the lowest values (–1 ℃) attained close to the coast (between 0 and 7 km) (Figures 6c–d).

The largest variability in the offshore waters was associated with the thermocline depth (temperature anomaly of about –1.0 ℃). In spring, the thermocline was almost horizontal and located at ~120 m depth. In summer, the thermocline was located higher in the water column (~70 m depth), with a slight inclination (deeper in the offshore). In autumn, the thermocline depth increased to 100 m, with a more pronounced inclination. The thermocline inclination in summer and autumn was most likely due to upwelling processes when the winds were upwelling-favourable (Figure 2). In winter, the thermocline was absent because the Leeuwin current dominated the offshore waters.

*Salinity anomaly*

In summer and autumn, the salinity anomaly was higher on the upper shelf than in the offshore waters, mainly because of evaporation (Figures 7b, c). A cross-shelf salinity gradient was also present. The salinity was more uniform in the surface waters offshore. In spring, high salinity water was present at >100 m corresponding to the colder water (Figure 6a). In winter, salinity gradients were absent along the whole transect (Figure 7d).

*Chlorophyll concentration anomaly*

The chlorophyll concentration anomalies revealed there were seasonal variations in the upper shelf and offshore waters (Figure 8). Across the whole transect, high chlorophyll concentration anomalies were present in the subsurface waters (i.e. not at the surface) and along the upper shelf. In spring, the highest chlorophyll concentration anomaly was found at the shelf break (Figure 8a) and was related to a horizontal temperature gradient (Figure 6a). A subsurface chlorophyll maximum (SCM), which extended over ~100 m depth, was present in the offshore waters. This SCM was associated with the temperature and salinity distribution (Figures 6a and 7a). In summer, the SCM was concentrated over a smaller depth range (< 50 m) in the offshore waters. On the continental slope, seaward of the shelf break (an area of 20–30 km), the chlorophyll concentration anomalies were more diffuse, most likely because of the variation in the upwelling and the diurnal cycle (Chen et al., 2017). In autumn and winter, the SCM was absent, but the chlorophyll concentration anomalies were



higher on the upper continental shelf. The autumn chlorophyll distribution corresponded to the presence of DSWCs on the upper shelf (Figures 7c and 8c).

### 3.5    Depth-integrated mean variability

We used a seven-year dataset of ocean glider deployments to examine the interannual variability in the temperature, salinity, and chlorophyll concentrations. The water properties in the surface to 30 m depth were averaged to yield depth-mean values (Figure 9). The gliders traverse in a sawtooth pattern, and as the depth increases, the surfacing spacing increases; thus there were gaps in the data for the deeper waters. All the water properties showed seasonal changes, but in this section, we focus on the interannual variability.

The year-to-year summer temperature range was > 4 ℃: the temperature was < 20.1 ℃ in February 2010 and increased to a maximum of 24.4 ℃ in March 2011 and also in February 2012 (Figure 9a). This maximum temperature was associated with the persistent La Niña event. In winter, the year-to-year temperature range was > 3 ℃: the temperature was > 21.2 ℃ in 2011 and decreased to a minimum of 18.4 ℃ in 2012 and 2014.

The concurrent depth-averaged salinity time series showed the waters were less saline (34.9) in August 2011 than they were in other years (Figure 9b). In March 2015, the highest salinity value (~35.9) was measured on the upper shelf. The lowest value (~35.5) was attained in 2011 and was associated with the warmer water.

The depth-averaged chlorophyll concentrations also had strong interannual variation (Figure 9c), with values ranging from 0.81 mg/m³ in May 2009 and 2010 to 1.8 mg/m³ in May 2014. The largest range in the chlorophyll concentrations was from March 2011 (chlorophyll concentration of 0.88 mg/m³) to March 2014 (chlorophyll concentration of 0.14 mg/m³). The lowest value of 0.18 mg/m³ was also recorded in March 2013.

Time series of the depth-averaged (surface 30 m) temperature, salinity, and chlorophyll measured 10 km from the coastline revealed strong seasonal and interannual variability, especially in response to El Niño and La Niña events (Figure 10). Figure 10 includes the same information as shown in Figure 9, except that variations at a single point (10 km) are shown as a time series. The seasonal cycle (Section 3.3; Figure 5a) indicated warmer, saltier water was present in summer and cooler, less saline water in winter.

In 2009/10, a moderate El Niño event occurred, which resulted in lower temperatures and higher salinity during the first half of 2010. The El Niño weakened the Leeuwin current, which entrained cooler, saltier water into the region from offshore (e.g. Woo and Pattiaratchi, 2008). The chlorophyll values were ~1 mg/m³, with a slight elevation in winter due to the seasonal bloom (Figure 10).

The 2009/10 El Niño was followed by a strong, extended La Niña between 2011 and 2014. The ocean glider data (Figure 10) captured several El Niño and La Niña effects on the water column properties: (1) a maximum temperature (> 24 ℃) was recorded in February 2011, which was an increase of > 4 ℃ from 2010. From 2011 to 2014, summer temperatures decreased; (2) a significant drop in salinity (> 0.5) occurred from 2010 to 2011. This salinity decrease was mainly due to a stronger Leeuwin current transporting lower salinity water into the





region. From 2011 to 2014, the salinity increased; (3) chlorophyll decreased from ~1 mg/m³ in 2011 to < 0.25 mg/m³ in early 2014 and then increased in May 2014.

### 3.6    Temperature, salinity, and chlorophyll during a storm event

The ocean glider data obtained from 17 to 28 April 2013 revealed a storm caused vertical mixing in the water
column and transported high chlorophyll water from the SCM to the surface. The first two transects (17–20
April 2013) were collected under low wind (< 5 m/s) conditions (Figure 11). The water column was vertically
stratified because of the presence of a DSWC on the upper shelf and a thermocline in the offshore waters
(Figures 12a,b). On the upper shelf, higher chlorophyll water was present in the DSWC's bottom layer. In the
deeper waters, the higher chlorophyll water was associated with the SCM (Figures 12m, n).

On 20 April, the winds increased to > 10 m/s. The winds were initially southerly and then changed to westerly
and continued to 23 April (Figure 11) causing vertical mixing of the water column. On the upper shelf, the
DSWC was eroded such that by 25 April, the temperature and salinity were vertically mixed in the water
column (Figures 12e, k). On 28 April, when the winds decreased, the salinity on the upper shelf was vertically
stratified (Figure 12l).

The winds and the vertical stratification also affected the chlorophyll distribution. Initially, high chlorophyll
concentrations were found in the DSWC on the upper shelf close to the seabed and in the SCM in the offshore
waters (Figure 12m). As the winds increased, the chlorophyll concentration became uniform through the water
column across the whole transect (Figure 12q). Note that the wind, although not strong (~10 m/s), was able to
mix the water column to ~80-m depth in the offshore waters and erode the thermocline and thus the SCM. The
SCM likely reformed (Figure 12s); however, with reduced solar heating and convective cooling, the
stratification would have weakened, which would have led to a well-mixed water column in late autumn and
early winter.





### 4. Discussion

In this paper, simultaneous water column data of ocean properties (temperature, salinity and chlorophyll fluorescence) together with meteorological data were used to examine the seasonal and inter-annual variability along the Rottnest continental shelf. Acquisition of multi-year sustained ocean observations using shipborne sampling is difficult (relative cost and weather dependence) and thus many studies have used satellite remote sensing data of sea surface temperature and ocean colour to determine season and inter-annual variability on continental shelves (e.g. Nieto and Melin, 2017; Kilpatrick et al., 2018). However, satellite derived data only provide information on the spatial variability on the surface of the ocean and thus variability in the sub-surface is unknown. Autonomous ocean gliders provide data from the sub-surface and have been used as a platform to collect multi-year sustained observations from the coastal ocean in mid-Atlantic Bight (Schofield et al., 2008). In this paper, ocean glider data collected along a single transect along the RCS shelf over the period 2009-2015 are presented. The data indicated distinct seasonal and interannual variation in temperature, salinity and chlorophyll concentrations. The chlorophyll variability was related to the changes in the temperature, salinity distribution that was related to changes in the physical forcing: (1) the local wind field; (2) the Leeuwin current system; and, (3) air–sea fluxes, especially in terms of surface cooling and evaporation. Due to the low tidal range (~0.6 m; Pattiaratchi and Eliot, 2009), tidal effects in the region were minimal. Previous studies in the region undertaken using remotely sensed imagery and limited shipborne observations have highlighted the seasonal variability (Lourey et al., 2006; Pearce et al., 2006; Fearns et al., 2007; Koslow et al., 2008; Pattiaratchi et al., 2011).

The seasonal wind regime in the region can be divided into three regimes (Pattiaratchi et al., 2011): (1) spring and summer (September–February)—strong, daily sea breezes dominate, with southerly winds often exceeding 15 m/s; (2) autumn (March–May)—the transition from the summer to the winter regimes occurs, and wind speeds are usually low; and (3) winter (June–August)—storms occur frequently.

The study region is located at 32 °S, close to the critical latitude (30 °S), where the inertial period is 24 hours. Because the diurnal sea breeze system also has a ~24 hour period, resonance occurs, which generates near-inertial waves (Mihanović et al., 2016; Chen, 2018). Field measurements revealed that near-inertial waves force the thermocline to oscillate at a diurnal timescale, with a vertical excursion > 50 m (Mihanović et al., 2016; Chen, 2018). This vertical excursion of the thermocline causes the thermocline to migrate along the continental slope on a diurnal timescale. This process has a strong influence on the sub-surface chlorophyll maximum (SCM) interaction at the continental slope: the chlorophyll anomaly indicates higher concentrations along the slope between water depths 50 m (shelf break) and 120 m (Figure 8b).

The main oceanic forcing in the region is the Leeuwin current, which flows along the 200-m depth contour and transports warm, low salinity water southward. The Leeuwin current is weakest in October, begins to accelerate in April, and reaches maximum speeds in June. The Leeuwin current transport decreases from July to October and then increases over summer (Wijeratne et al., 2018).

The study region is located in a Mediterranean climate zone, with hot, dry summers and mild, wet winters. The



annual evaporation rate exceeds 2 m (Pattiaratchi et al., 2011). There are no major land-based freshwater inputs to the region. Although the Swan River discharges at Fremantle, its freshwater component is low because rainfall is low during summer and autumn, and the river discharge is mainly deflected south in winter. The combination of evaporation and cooling is such that in summer, coastal heating and evaporation result in a band

of warm, high salinity water close to the coast; in winter, the nearshore waters are cooler (through heat loss to the atmosphere) and less saline (Pattiaratchi et al., 2011).

The impacts of all the physical forcing at the seasonal scale were reflected in the temperature and salinity (T/S) distribution across the shelf. The (T/S) structure across the continental shelf showed the water was warmest on

the upper shelf between January and March whilst the salinity increased (Figure 5). The warming was due to high solar insolation and the higher salinity through evaporation. From March to August, both the temperature and salinity decreased. This temperature decrease was due to atmospheric heat loss despite the transport of warmer water into the region by the Leeuwin Current. The salinity decrease was due to the advection of lower salinity water from the Leeuwin current. From August to January, both the temperature and salinity increased

because of the increasing solar insolation and evaporation. The offshore waters showed a similar seasonal pattern.

In general, both inshore and offshore chlorophyll concentrations were higher in autumn and winter (March to August) than they were in spring and summer. Maximum values were attained in May for the inshore (1.1

mg/m$^3$) and offshore waters (0.85 mg/m$^3$). The chlorophyll difference between summer and winter inshore (0.75 mg/m$^3$) was larger than it was offshore (0.45 mg/m$^3$). A similar seasonal pattern was found in studies conducted over the past two decades. Pearce et al. (2000) found offshore chlorophyll concentrations between 1979 and 1986 were highest between May and August. Fearns et al. (2007) found a clear seasonal cycle, with maximum values attained between May and July from 1997 to 2004.

The main differences between spring/summer and autumn/winter were found in the water column structure, especially in the offshore regions. The offshore waters were vertically stratified in spring and summer and vertically mixed in autumn and winter (Figures 6–8). The pycnocline in spring and summer initiated the SCM. Koslow et al. (2008) observed the summer SCM in a layer above the nutricline at 100 m depth when the water

column was stratified. The nearshore autumn bloom coincided with the DSWCs, which regularly occur in autumn (Pattiaratchi et al., 2011). In winter, high chlorophyll concentrations were uniformly distributed inshore because of winter cooling and storm-induced mixing of the water column (Longhurst, 2007; Koslow et al., 2008; Chen et al., 2017).

Our results are in broad agreement with those of Koslow et al. (2008), who used ship-based sampling data and satellite remote sensing data to study phytoplankton in the same region. Koslow et al. (2008) found that: (1) the primary productivity and chlorophyll concentrations were lower offshore in summer when the water column was stratified and most of the chlorophyll was contained in the SCM; (2) phytoplankton blooms in late autumn and winter coincided with the period when the Leeuwin current flow was strongest, and the winter bloom was

due to cooling and storms, which promoted mixing of the upper water column. We also observed higher autumn





and winter chlorophyll concentrations and a vertically mixed upper water column in the offshore region.

In summary, the observed patterns of seasonal variability in the chlorophyll concentrations were related to the changes in the water's physical properties, which were affected by the seasonally changing physical forcing. These findings were similar to those from Vidal et al.'s (2017) study of the Iberian continental shelf, which is a region with similar dynamics (e.g. eastern poleward boundary current, upwelling-favourable winds) to the Rottnest continental shelf. Vidal et al. found that the physical forcing's frequency and intensity affected the chlorophyll variability.

In spring and summer, the offshore waters were vertically stratified, with a surface mixed layer and a well-defined pycnocline; the Leeuwin current was weak and located farther offshore (Pearce and Pattiaratchi, 1999). The strong, sustained winds mixed the upper water column to > 50 m depth. The pycnocline prevented nutrients moving from beneath the pycnocline; however, the high light penetrating into this region allowed the SCM to form. At the shelf break, the pycnocline moving along the slope and on the upper shelf caused higher chlorophyll concentrations at the shelf break and in the bottom layer. In spring, the higher chlorophyll concentrations at the shelf break were located where there were temperature gradients between the upper shelf and the offshore waters (i.e. a shelf break thermal front was present; Figures 6 and 8).

### 4.1 Interannual variability

The ocean glider dataset collected between 2009 and 2015 revealed strong interannual variability in the region (Figure 10). Local and basin-scale ocean forcing affected the coastal hydrography (temperature and salinity) and biological variables (chlorophyll). Pearce and Feng (2013) analysed large-scale (monthly), satellite-derived, sea surface temperature data and found that coastal water temperatures off south-west Australia varied interannually and were linked to the ENSO cycle. During La Niña events, a strong Leeuwin current transported warm water southwards (Pattiaratchi and Buchan, 1988; Pearce and Philips, 1988; Feng et al., 2003, 2008), whereas during El Niño events, the Leeuwin current was weaker with generally lower water temperatures (Pattiaratchi and Buchan, 1988; Pearce et al., 2006). Several ENSO events occurred during the study period (http://www.bom.gov.au/climate/enso/lnlist/): (1) the 2009–2010 El Niño; (2) the 2010–2013 La Niña; and (3) 2014–2015 neutral conditions.

The study region experienced a 'marine heat wave' in February and March 2011, which was related to a La Niña event and defined as Ningaloo Niña (Feng et al., 2013). This event increased the Leeuwin current's volume transport in February—an unusual event at this time of the year—and caused high sea surface temperature anomalies (~5 ℃ higher than normal) off Australia's west coast (Feng et al., 2013). It affected extensions and contractions in species distributions and variations in recruitment and growth rates of species, and caused coral bleaching and the mass death of marine life, with short-term and long-term impacts (Pearce et al., 2011). The glider data revealed that extreme depth-integrated (upper 30 m) temperatures (up to 3.5 ℃ above average) occurred in March 2011 and February 2012 (Figure 9).





The glider data captured the transition between the 2009–2010 El Niño and the extended 2011–2014 La Niña (Figure 10). Large changes in the temperature (increase > 4 ℃) and salinity (decrease by 0.5) occurred between the El Niño and the La Niña events. The temperature decreased and the salinity increased over the same period, with an accompanying decrease in the chlorophyll concentration from ~1 mg/m³ in 2011 to < 0.25 mg/m³ in early 2014. A strong Leeuwin current, which transported warmer, low salinity, low nutrient water from the north to the south, affected the interannual variability in the temperature and salinity and most likely the chlorophyll concentration. A small decrease in the number of winter storms between 2011 and 2014 (Wandres et al., 2017) might also have reduced local recycling of nutrients (Chen et al., 2017).

The extended impacts of the heat wave were also found in temperate reef communities in Western Australia, with the loss of kelp forests, which were replaced by seaweed turfs (Wernberg et al., 2016). Here the marine heat waves forced the contraction of a 100-km area of extensive kelp forests and the replacement of temperate species in the reefs by species characteristic of subtropical and tropical waters. Wernberg et al. (2016) reported that the heat wave effects persisted for many years, and almost five years after the heat wave, the kelp forests had not recovered. The decrease in the chlorophyll concentrations between 2011 and 2014 showed the heat wave affected the pelagic and benthic ecosystems over an extended period.

## 5. Conclusions

A seven-year, high-resolution ocean glider dataset was used to study the seasonal and interannual variability across the Rottnest continental shelf and indicated that the temperature, salinity, and chlorophyll concentrations had a strong, seasonal and interannual variability. Controlling influences for the seasonal variability were from changes in the wind forcing, dense shelf water cascades (DSWC) and the Leeuwin Current scale, boundary and shelf current systems whilst interannual variability was through El Niño Southern Oscillation (ENSO) events. In spring and summer, the shallow waters were well mixed through wind mixing and the deeper waters (> 50 m) were vertically stratified in temperature that contributed to the formation of a subsurface chlorophyll maximum (SCM). With the onset of storms in late autumn, the water column was well mixed with the SCM absent. On the inner shelf, chlorophyll concentrations were highest in autumn and winter; DSWCs were also the main physical feature during autumn and winter. Chlorophyll concentration was higher closer to the sea bed than at the surface in spring, summer, and autumn. The seasonal patterns coincided with changes in the wind field (weaker winds in autumn) and air–sea fluxes (winter cooling and summer evaporation). Inter-annual variation was associated with ENSO events. Lower temperatures, higher salinity, and higher chlorophyll fluorescence were associated with the El Niño event in 2010. During the strong La Niña event in 2011, temperatures increased and salinity and chlorophyll fluorescence decreased. These changes were mainly associated with an increase in the strength of the Leeuwin current that transported warmer, lower salinity, low nutrient water into the region. Over subsequent years, the temperatures gradually decreased, the salinity increased, and the chlorophyll continued to decrease. In the autumn of 2014, the chlorophyll increased. It is concluded that the observed seasonal and inter-annual variability in chlorophyll concentrations were related to the changes in physical forcing.



**Acknowledgments and data**

The ocean glider data used in this paper were collected as part of the Integrated Marine Observing System (IMOS) by the Australian National Facility for Ocean Gliders (ANFOG) located at the University of Western Australia. The IMOS is supported by the Australian Government through the National Collaborative Research Infrastructure Strategy. The Bureau of Meteorology obtained the meteorological data. Funding for this study

5  was provided by International Postgraduate Research Scholarship, Australian Postgraduate Award and University Postgraduate Award. The authors acknowledge the support of Dennis Stanley, Paul Thompson, Kah Kiat Hong, and Mun Woo in the tasks associated with the deployment, recovery, piloting, and data QA/QC of the ocean gliders. All the ocean glider data are available through the Australian Ocean Data Network (https://portal.aodn.org.au).



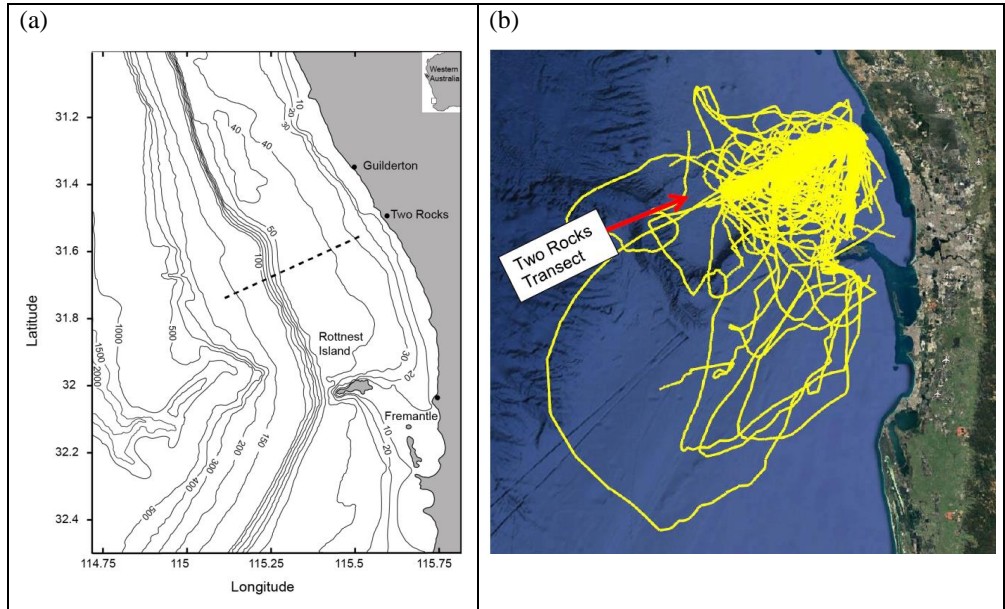

**Figure 1: (a) The study area. The dashed line denotes the location of the glider transect. Bathymetric contours are in metres; (b) tracks of all the Slocum ocean glider deployments from 2009 to 2015.**




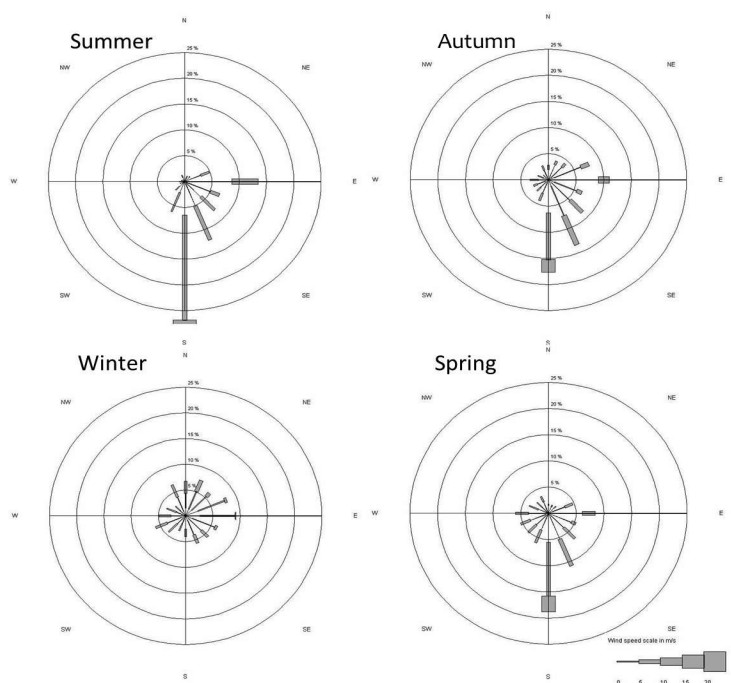

Figure 2: Seasonal wind rose climatology for 2010–2014.





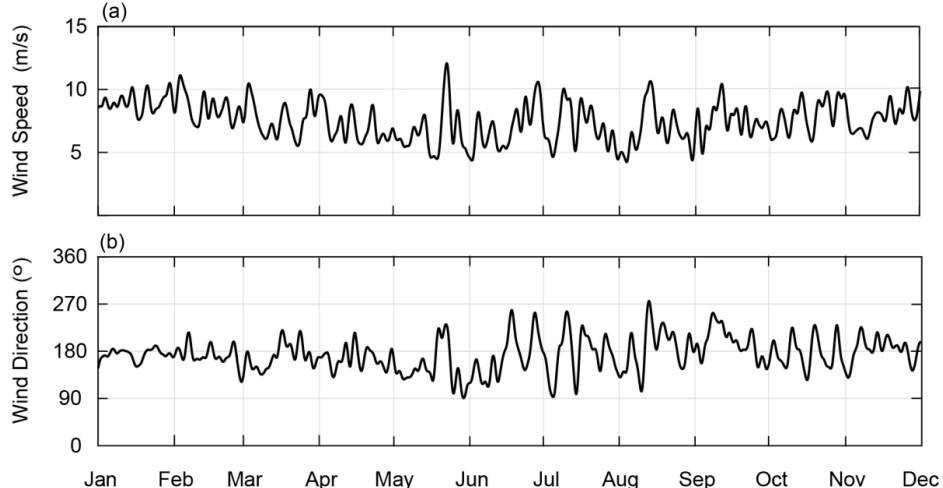

**Figure 3: Time series of the (a) mean daily wind speeds and (b) wind direction in 2010 recorded at the Rottnest Island meteorological station (Figure 1).**



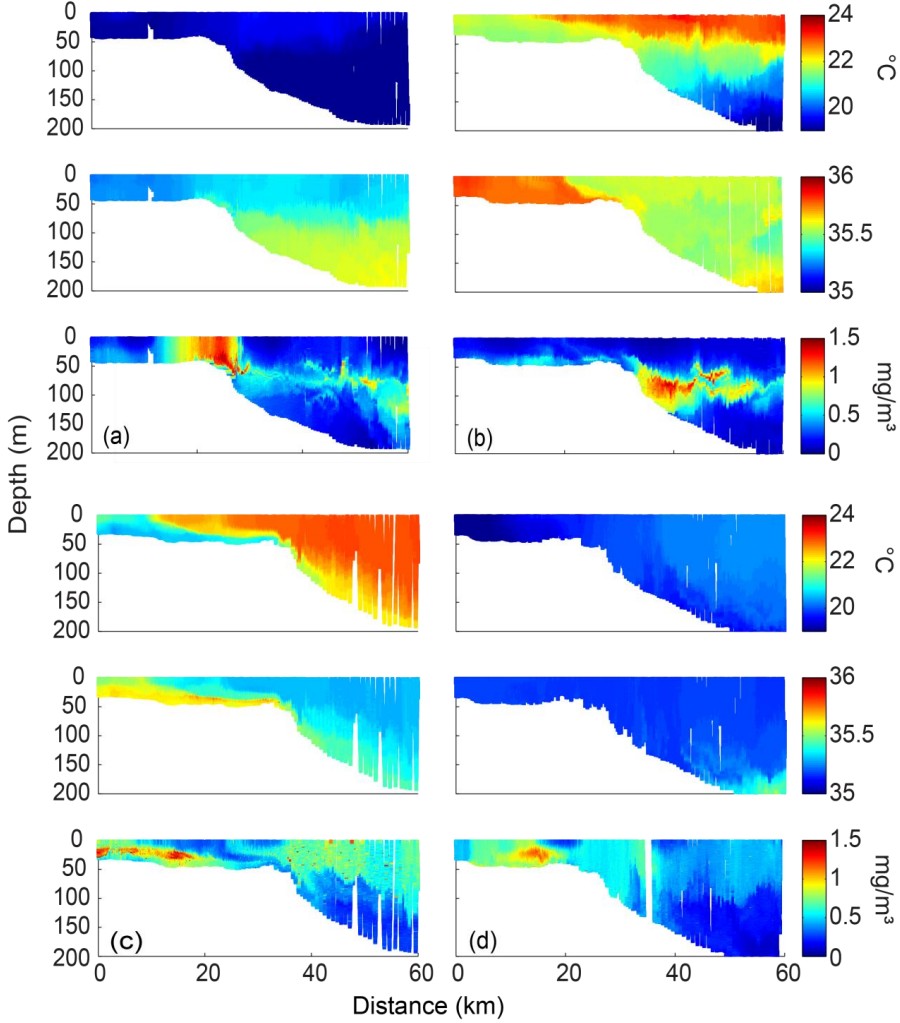

**Figure 4: Cross-shelf transects of temperature (°C), salinity, and chlorophyll (mg/m³) obtained along the Two Rocks transect in (a) spring (21–23 October 2013); (b) summer (28 February–3 March 2014); (c) autumn (18–21 May 2009); and (d) winter (9–11 August 2012).**




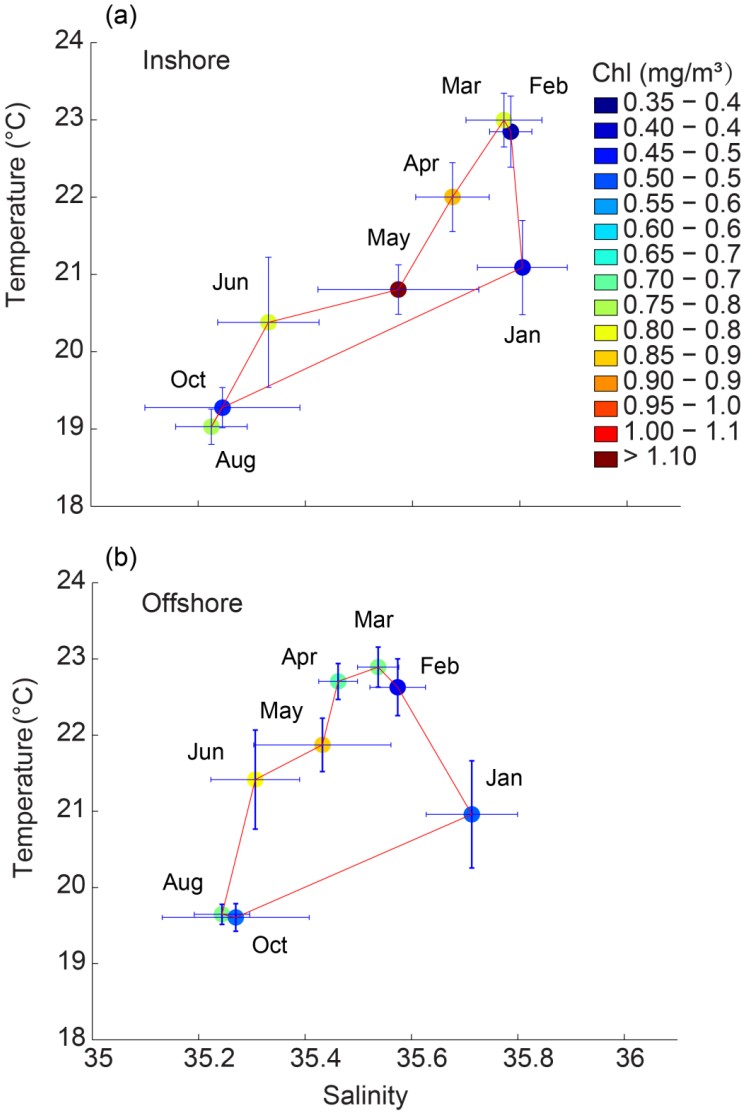

**Figure 5: Monthly averaged temperature–salinity diagram with the chlorophyll (Chl) values (mg/m³) for the Two Rocks transect between 2009 and 2015. The horizontal error bars indicate the standard deviation of salinity, and the vertical error bars indicate the standard deviation of temperature.**



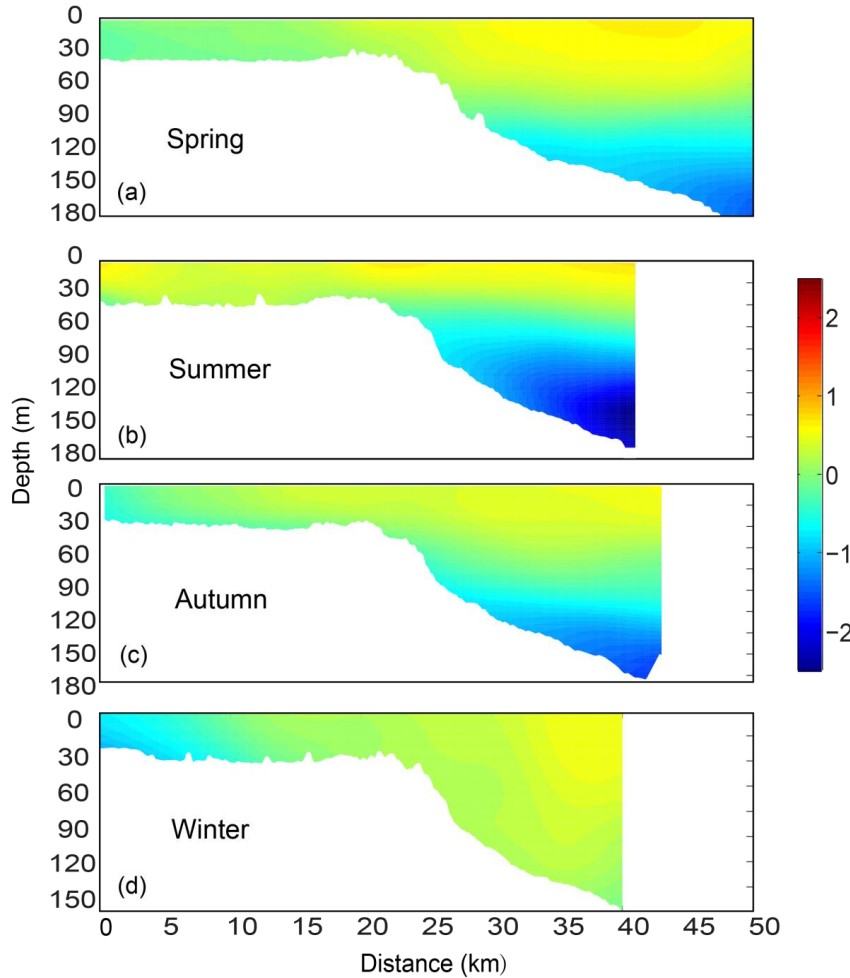

**Figure 6: Mean vertical structure anomaly of the temperatures (°C) in (a) spring, (b) summer, (c) autumn, and (d) winter, averaged seasonally over distance and depth across the Rottnest continental shelf between 2009 and 2015.**





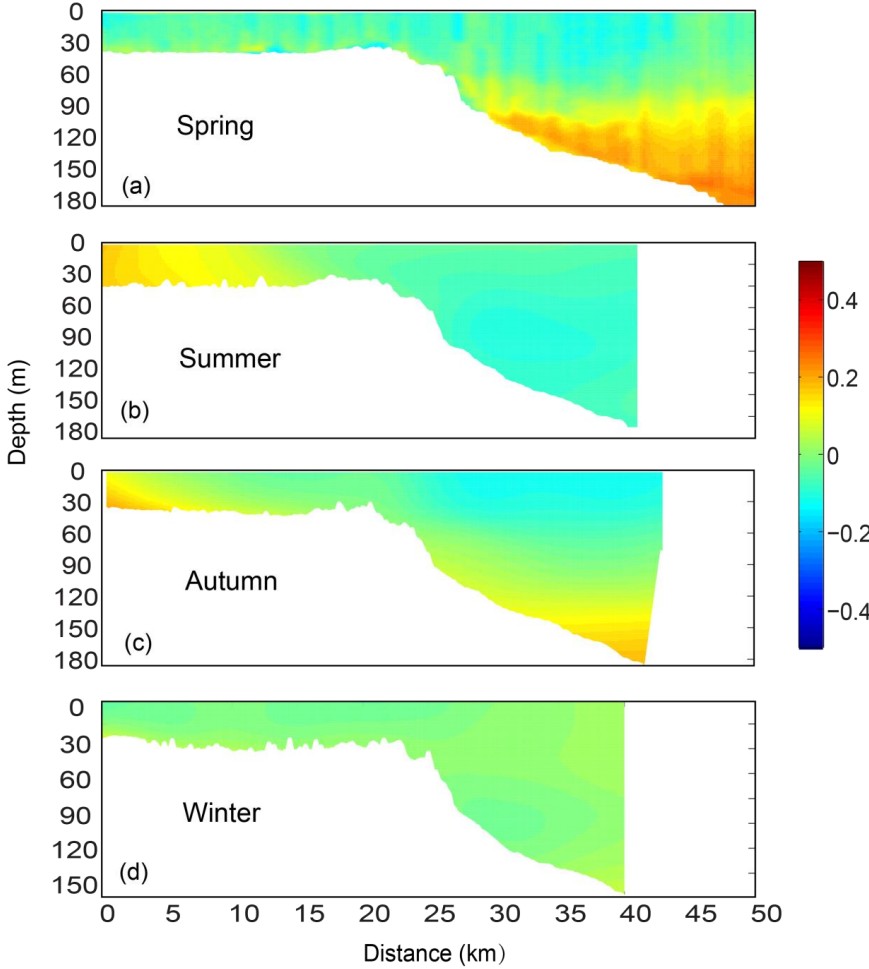

**Figure 7: Mean vertical structure anomaly of the salinity in (a) spring, (b) summer, (c) autumn, and (d) winter, averaged seasonally over distance and depth across the Rottnest continental shelf between 2009 and 2015.**




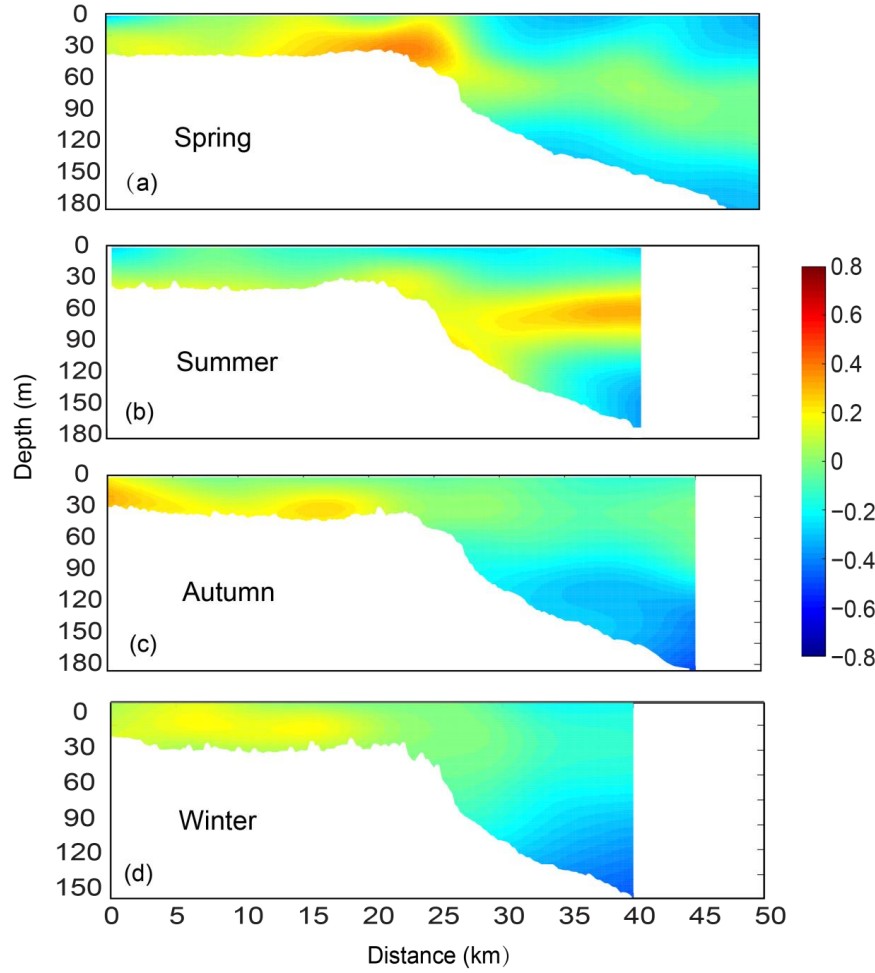

**Figure 8: Mean vertical structure anomaly of the fluorescence** $(\mathrm{mg/m^3})$ **in (a) spring, (b) summer, (c) autumn, and (d) winter, averaged seasonally over distance and depth across the Rottnest continental shelf between 2009 and 2015.**





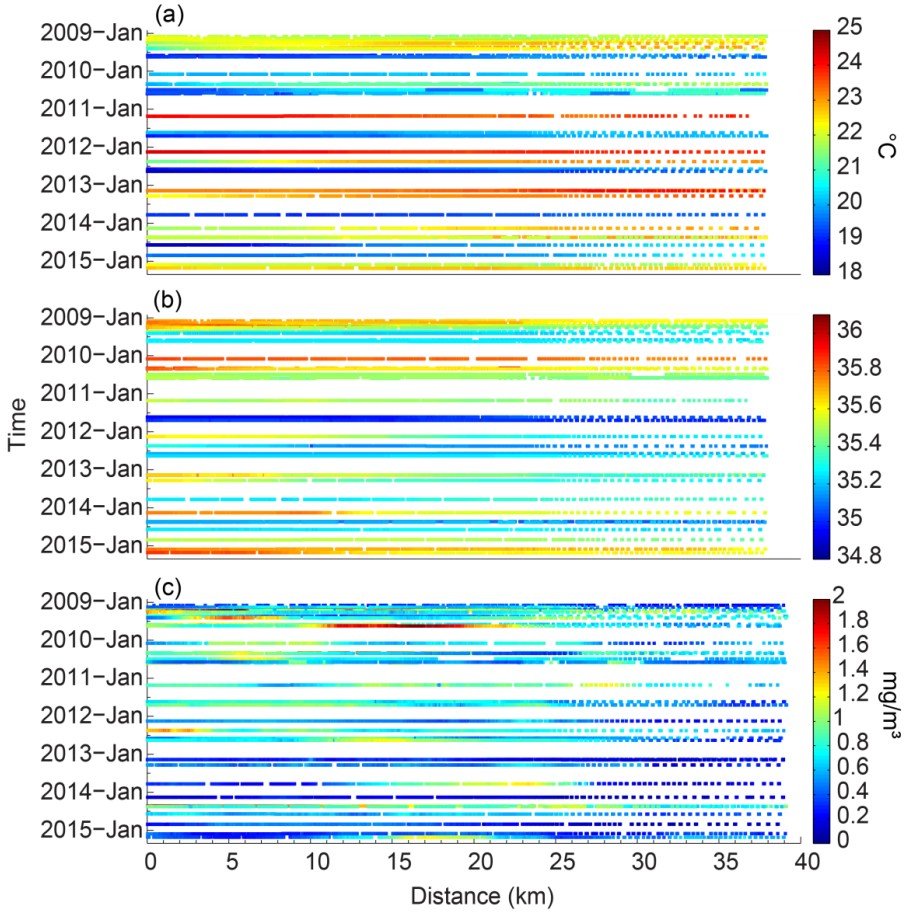

**Figure 9: Time–distance series of the (a) temperature (℃), (b) salinity, and (c) chlorophyll (mg/m³), averaged for the top 30 m of water along the Rottnest continental shelf between January 2009 and March 2015. Zero distance (0 km) denotes the start of the glider transect.**





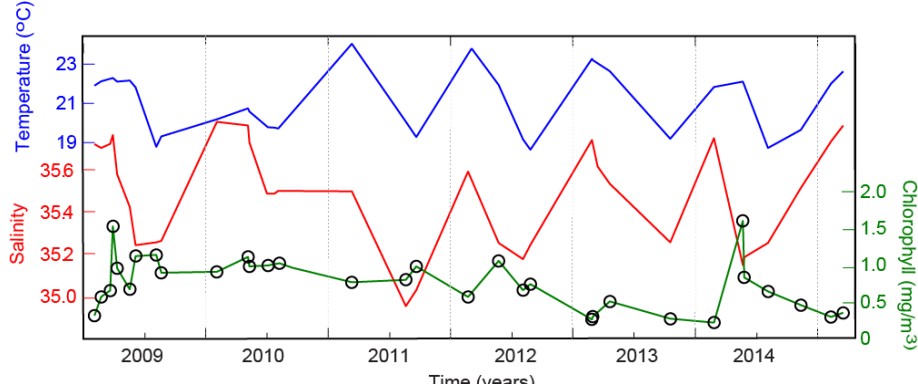

**Figure 10: Time series of the depth-averaged temperature, salinity, and chlorophyll 10 km from the start of the glider transects.**





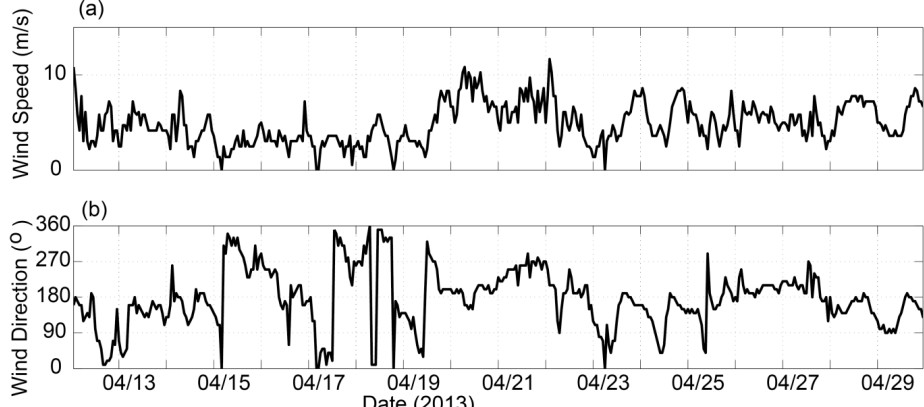

**Figure 11: (a) Wind speed (m/s) and (b) wind direction (°) along the Rottnest continental shelf between 12 and 30 April 2013.**




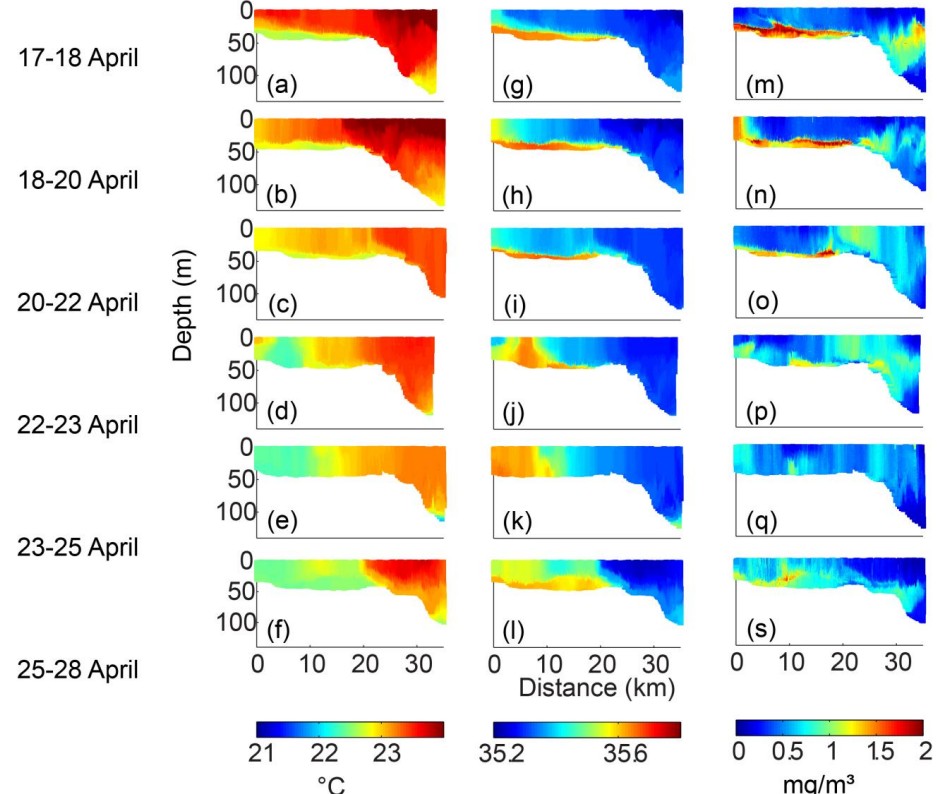

**Figure 12: Vertical cross-sections of (a–f) temperature (°C), (g–l) salinity, and (m–s) chlorophyll (mg/m³) across the Rottnest continental shelf between 17 and 28 April 2013 obtained with the ocean glider.**



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
