# Peer review of "Seasonal and interannual variability of water column properties along the Rottnest continental shelf, south-west Australia"

_Ocean Science, 2018_

## Referee Comment (RC1) · J. Kämpf (Referee) · 6 Nov 2018

This is a well-written scientific report of the seasonal and interannual variability of water column properties along the Rottnest continental shelf, south-west Australia, based on glider data. The study includes chlorophyll fluorescence which adds relevance to carbon cycles and marine ecosystems. Here are a few comments that the authors should address in their revisions.

1) The abstract is tooooooooo long. Can it be shortened?

2) Colors can be deceptive. Please add contours to figures 6,7 and 8.

[Figure]

3) What do you mean by structure anomalies in Figures 6,7, and 8? Anomalies relative to a surface value? Relative to a seasonal average? Or an annual average? Please show absolute distributions or, at least, the reference value/profile that your anomalies are based on. Perhaps, you should also present seawater density distributions and discuss seasonal variations of the density structure.

4) Figure 9 is difficult to interpret. Is there a way to fill the data gaps using satellite SST? Why do you present this figure? Perhaps this would be better placed in the methodology section together with a discussion of data gaps?

5) It would be nice to have true chl-a values rather than just data from the BBFL2SLO optical sensor. How confident are you that your fluorescence data represent true chl-a, in particular close to the seafloor? How is this bottom chl-a maximum created? Is there any reason why you decided not to discuss CDOM?

6) In our previous study (Kämpf and Kavi, 2017), we identified seasonal chl-a maxima in the Great Australian Bight in austral autumn months. Is this feature, which is not too far away from your study region, consistent with your observations? If so, please discuss this.

7) In the last sentence of the abstract you claim that "It is concluded that the observed seasonal and inter-annual variability in chlorophyll fluorescence concentrations were related to the changes in physical forcing (wind forcing, Leeuwin Current and air-sea fluxes)." This statement is far too general and misleading given that you didn't analyze air-sea fluxes. You also don't specify what type of air-sea flux you are referring to. Dust influences? Heat fluxes? Neither did you calculate the classical upwelling index or estimate the possible influence of mesoscale eddies that could lead to dynamic uplift of nutrient-rich water across the shelf break or passing baroclinic coastally trapped waves.... Much more effort would be required to identify reasons of the observed variability of chlorophyll fluorescence concentrations.

8) In the autumn of 2014, the chlorophyll fluorescence increased (> 1 mg m–3). Do

you know why?

Reference: Kämpf, J., and A. Kavi (2017), On the "hidden" phytoplankton blooms on Australia's southern shelves, Geophys. Res. Lett., 44, 1466–1473, doi: 10.1002/2016GL072096.

---

## Referee Comment (RC2) · Anonymous Referee #2 · 6 Nov 2018

Abstract Sentences 2 and 3 needs integration. They can be mix together integrating the information to report. Introduction Line 2. I would prefer to start the sentence like. "Among phytoplankton pigments, chlorophyll. . ..." There are other pigments in phytoplankton. I think that the is chlorophyll a that was used as indicator of phytoplankton biomass or you use the total amount of chlorophyll? Depending of your answer change the sentence accordingly. On paragraph 25 and 30, the sentence starting by "It is major is a mechanism. . ." I think this is a typing error remove the second "is" and "a". Page 5 paragraph 10. Please remove "highly" from the sentence Thus the RCA is a highly nutrient". . . Page 5 paragraph 20. ". . .are not weather limited." The gliders were able to fly even during bad weather conditions and strong winds? If so why you mentioned that

in some seasons data were not present and the graphs have gaps? Methods Page 6. Paragraph 10. Of course, this methodology is a huge advance that regular measurements performed but I was also wondering if two or three days took to complete the transect was not too much time. In very dynamic areas, like upwelling areas, you might have complete different conditions within the 3 days for adjacent areas. Page 6 paraph 20. Did you performed any intercalibration exercise between the data collected from the chlorophyll fluorescence recorded and the quantification of chlorophyll (mg/m3). Again total chlorophyll or chlorophyll a only. Page 6. Last paragraph is very confusing. I don't understand if you reach the conclusion that the data was anormal by subtracting to the seasonal mean. You say previously that you perform quality control on data. Why you don't exclude the anormal data there....so you won't have to deal with them again later. I think that you should try to rephrase and clarify that paragraph. Results Page 8. 3.3 Methods section; first line. This information must be in Methods section and justified why there are no mean for the season for that months, if you claim that gathering of data using gliders are not weather limited. Page 9 Paragraph 25 Typing error. Replace 6a by 7a. Pag 10. The 3.5 section of results were very difficult to follow because the figure 9 were not understandable and I advise rebuilding it in clearer manner. Probably because I was confused with the figure I think that the paragraph 20 description was not correct. You have a higher chlorophyll concentration than 0.81 mg/m3 mentioned for 2009, between 10 and 20 m depth with chlorophyll concentration ranging 1.8mg/m3. Figure 10 was very good evidencing the different pattern between El Niño e La Niña. Discussion Page 12. Line 14 (I think) Typing error : repeated. Paragraph 20 to 25 must be in introduction. Paragraph 35 to 40 was already described in introduction. Paragraph 25 and 35 must integrated with the obtained data by proving examples of the physical processes I think that your discussion must be improved by comparing your data with another data from upwelling coastal areas also impacted by El Niño and La Niña and compared the impact results in terms of chlorophyll and consequently in productivity between areas or with former events. It is very important to bring awareness of climate change and the huge effects they have in coastal dynamics

and phytoplankton biomass and overall productivity giving relevant to studies like the ones you developed.

---

## Author Comment (AC1) · 19 Jan 2019

Response to reviewers: os-2018-115 (Submitted to Ocean Science)

Seasonal and inter-annual variability of water column properties along the Rottnest continental shelf, south-west Australia by Miaoju Chen, Charitha Pattiaratchi, Anas Ghadouani and Christine Hanson.

We would like to thank and acknowledge both reviewers and the editor for their careful reading and constructive comments on the manuscript. There were no public comments. We believe that we have addressed the issues raised by reviewers and the

proposed changes to the manuscript are detailed in this document. We trust that the reviewers and the editor will find that the suggested changes will make the manuscript to be suitable for publication.

In the following, black indicates the comments by the reviewer; blue is our response to the reviewers. The text in red are suggested changes to the manuscript.

Reviewer #1

(1) The abstract is toooooooo long. Can it be shortened?

The journal guidelines does not specify a word limit for the abstract and currently consist of 360 words. At the suggestion of the referee we will revise and shorten the abstract

(2) Colors can be deceptive. Please add contours to Figures 6,7 and 8.

At the suggestion of the referee we have included contours in Figures 6, 7 and 8. As an example, the temperature distribution for spring is provided below:

(3) What do you mean by structure anomalies in Figures 6,7, and 8? Anomalies relative to a surface value? Relative to a seasonal average? Or an annual average? Please show absolute distributions or, at least, the reference value/profile that your anomalies are based on. Perhaps, you should also present seawater density distributions and discuss seasonal variations of the density structure.

Thank you for your comments. We first examined the mean values by season. However, the seasonal variation in parameters obscured the patters and thus we presented anomalies that were calculated relative to the seasonal mean over the measurement period calculated though water depth and distance. For reference we have included absolute distributions as well as density distributions as supplementary information. The mean distribution for each parameter for the different seasons are included as a Table.
We have also modified the text to make clearer how the anomalies were calculated with the following text with a new Table added to indicate the value for the mean for each parameter over different seasons.

The text has been modified as follows:

When examining the seasonal changes it was found that the changes in the mean values obscured the seasonal variability of each parameter (temperature, salinity, and chlorophyll). Hence, in addition to presenting the measured values we also calculated anomalies to remove the influence of the seasonal variability. The procedure for each parameter (∼28 million individual points) was as follows: (1) data were interpolated onto a common grid across the cross shelf transect; (2) transects were then sorted according to season: spring (September-November), summer (December-February), autumn (March-May) and winter (June-August); (3) the mean value across the whole transect (i.e. through water depth and across distance) for each season was calculated (Table 1); and, (4) the anomaly at each grid point was calculated by subtracting the seasonal mean from values at each point.

Table 1 – Mean values of temperature. Salinity and chlorophyll fluorescence for each season used to calculate the anomalies. Temperature (°Celsius) Salinity Chlorophyll fluorescence (mg/m$^3$) Spring 19.4 35.35 0.49 Summer 21.9 35.61 0.46 Autumn 22.4 35.42 0.71 Winter 19.8 35.27 0.68

  (4) Figure 9 is difficult to interpret. Is there a way to fill the data gaps using satellite SST? Why do you present this figure? Perhaps this would be better placed in the methodology section together with a discussion of data gaps?

It is not possible to fill the data gaps using satellite imagery (for SST or chlorophyll) as these are not surface values – rather they are depth integrated (surface 30 m) values. They are also time averaged – each line represent a single glider deployment lasting 3-4 weeks. In the methods section we have noted that as the glider moved in a saw tooth pattern, and gaps in the data occur when sampling deep waters – resulting in

data gaps in the deeper waters.

We have retained this figure as we believe that it illustrates the inter-annual variability. For example it highlights the marine heat wave (red lines - summer 2011) and cooler summers (yellow line – summer 2009, 2015). Similarly the figures indicates higher salinity during these summers (red line - summers 2010, 2014, 2015). We have highlighted these in the revised manuscript.

(5) It would be nice to have true chl-a values rather than just data from the BBFL2SLO optical sensor. How confident are you that your fluorescence data represent true chl-a, in particular close to the seafloor? How is this bottom chl-a maximum created? Is there any reason why you decided not to discuss CDOM?

We agree with the reviewer that 'true' chlorophyll values will add great value to this study, and we are aware that a common practice for compensating for the variability in fluorescence yield is to calibrate a fluorometer through the statistical comparison of fluorescence readings with measurements of concentration of chlorophyll from concurrently collected water sample (Cullen and Lewis 19956; Hersh and Leo 2012). However, due to the nature of glider deployments, which operate for extended periods of time and space without human interaction, marinating a water sampling regime is neither logistically nor financially feasible.

Should be noted that few studies have used 'true' chlorophyll a to define seasonal and inter-annual variability through the water column. The collection of routine water samples (say for HPLC or acetone extractions) for long time period are often not possible due to operational and financial considerations. Similar studies use satellite derived chlorophyll which is an indirect measurement relating upwelling radiance to chlorophyll a but is also limited mainly to surface values. Thus we believe that the data presented in this paper is unique.

However, as part of the IMOS ocean glider program we have undertaken many studies to address the conversion/relationship between the fluorescence values from the

BBFL2SLO optical sensor and 'true' Chlorophyll a. These were undertaken both in the laboratory (Earp et al., 2011) and in the field (Thomson et al., 2015). In the latter, we attached a glider to a rosette sampler and collected concurrent data from the glider and Niskin bottles at surface, mid-depth and bottom of the water column in 100m depth. The water samples were subjected to HPLC analyses to determine the 'true' Chlorophyll a concentrations. The comparison between ocean glider derived fluorescence and the HPLC Chlorophyll a concentrations was very good with r2 > 0.75 (n > 100) in the range 0.17 to 0.21 (mg m-3).

A recent study by Beck (2016) found that, through inter-comparison of chlorophyll a and Wetlabs ECOPUCK derived fluorescence on ocean gliders, the original manufacturer's recommendation for the estimation of chlorophyll a from fluorescence provided the best estimate.

The bottom chl-a maximum is created in many ways. The study region has very clear water and thus light penetrated is large (1% light level is > 100 m). The region is oligotrophic so there is no nutrients in the water column. Our shipborne measurements indicates that the nitrate concentrations were below detection levels (Twomey et al., 2007). We believe that there is some supply of nutrients onto the bottom layer through two possible sources: (1) regeneration from the organic matter on the seabed particularly during storm events; and, (2) advection onto the shelf from offshore through upwelling – this also may indicate the sub-surface chlorophyll maximum 'migrating' onto the shelf.

Yes there is a very good reason why we decided not to discuss CDOM – in a region with very little riverine input the CDOM concentrations were very small – almost negligible – except during occasional storm events. When averaged over a season there was no detectable changes. Similarly backscatter (a proxy for suspended matter). Hence, this paper is addressing the variability in chlorophyll concentrations only. A paper in preparation for publication is addressing the short-term changes of order days.

(6) In our previous study (Kämpf and Kavi, 2017), we identified seasonal chl-a maxima in the Great Australian Bight in austral autumn months. Is this feature, which is not too far away from your study region, consistent with your observations? If so, please discuss this.

Thank you for the comment. We have read through Kämpf and Kavi, (2017) and included this reference and discussed in section 4 as follows:

The observed surface chlorophyll features agreed with Kämpf and Kavi (2017), who showed widespread phytoplankton blooms (chlorophyll concentrations ∼1mg/m3) during autumn and winter using satellite data along the southern Australia coastline.

(7) In the last sentence of the abstract you claim that "It is concluded that the observed seasonal and inter-annual variability in chlorophyll fluorescence concentrations were related to the changes in physical forcing (wind forcing, Leeuwin Current and air-sea fluxes)." This statement is far too general and misleading given that you didn't analyze air-sea fluxes. You also don't specify what type of air-sea flux you are referring to. Dust influences? Heat fluxes? Neither did you calculate the classical upwelling index or estimate the possible influence of mesoscale eddies that could lead to dynamic uplift of nutrient-rich water across the shelf break or passing baroclinic coastally trapped waves.... Much more effort would be required to identify reasons of the observed variability of chlorophyll fluorescence concentrations.

We acknowledge that we have not fully elaborated on the changes to the physical forcing that contribute to the observed variability chlorophyll fluorescence concentrations. There are many different physical processes that contribute to this variability: the Reviewer has highlighted meso-scale eddies, coastally trapped waves as examples, others include diurnal upwelling and action of storms. However, all of these processes act over periods of order days or weeks. This study is concentrated on seasonal scales and higher – thus data have been averaged over a period of 3 months which does not allow for these processes to be identified – follow up publications will address diurnal

upwelling and impact of storm systems.

The Reviewer questions why the classical upwelling index was calculated. There are many reasons: 1. The paper is not based on upwelling. Upwelling favorable winds occur during spring/summer but the maximum chlorophyll occur in late autumn and winter. Thus chlorophyll fluorescence concentrations are is not only controlled by the wind and upwelling 2. The classical upwelling index is not applicable to this region due to the presence of the pressure gradient due to Leeuwin Current. This was addressed in a recent paper by Rossi et al. (2013) who applied an improved composite dynamical upwelling index that accounts for the role of alongshore pressure gradients counteracting the coastal Ekman divergence. The results indicated that upwelling was sporadic along the whole coast with the occurrence of transient upwelling events lasting 3–10 days changing in space and time. The study regions (at 31.5o) and consisted of up to 12 upwelling days per month during the austral spring/summer. The intensity of intermittent upwelling is influenced by the upwelling favourable winds, the characteristics of the Leeuwin Current and the local topography. As this study already exists there was no requirement to calculate the classical upwelling index in the paper. However, reference to Rossi et al. paper and its findings are included in the revised paper.

The physical forcing that influence chlorophyll concentrations are changes in wind forcing, Leeuwin Current and air-sea fluxes of heat and water. We have highlighted this in the discussion of the revised paper. We have indicated seasonal changes in each of these processes are: (1) strong southerly winds in spring/summer, weak in autumn and storms during winter; (2) LC being weak in summer and strong in winter; and, (3) evaporation dominance in summer and cooling in winter due to changes in air-sea fluxes of heat and water that leads to the formation of dense shelf water cascades in autumn and winter. We have referred to Pattiaratchi et al. (2011) paper that describes the seasonal cycle of air-sea fluxes and its influence on the continental shelf.

(8) In the autumn of 2014, the chlorophyll fluorescence increased (> 1 mg m–3). Do you know why?
Thank you for your comment. Yes we do have an explanation and it explains the peak during autumn 2009. The Leeuwin current is strongest in autumn and winter (mean transport: ∼5–6 Sv) and weaker during summer (mean transport: ∼2 Sv). A recent paper by Wijeratne et al., 2018) presented results of boundary current transport around Australia from a high resolution simulation over a 15 year period. The transport across a cross-shelf section at 31.5oS extending to the deeper ocean indicated that in January/February of 2009 and 2014 the southward mean monthly transport of the Leeuwin Current was very weak, < 0.5 Sv and close to zero. In contrast during the period 2010-2013 the monthly mean transport was mainly > 1.0 Sv. So how could lead to increased chlorophyll values – one explanation is that a reduction in Leeuwin Current would lead to a shallower mixed layer during the summer. When the winter storms arrive in late autumn the shallow mixed layer broken down more easily bringing nutrients onto the upper layer that allows for higher phytoplankton growth and thus higher chlorophyll. We highlight this process in Figure 12. We have also examined the number of major storms that impacted the study region over the period April-June with the following results: 2009: 7; 2010: 2 ; 2011: 5 ; 2012: 0; 2013: 5 ; 2014: 7. Thus over this period 2009 and 2014 had the more storms than other years, perhaps giving credence to this theory. We have included this explanation in the final manuscript.

References: Beck M. (2016), Defining a multi-parameter optics-based approach for estimating Chlorophyll a concentration using ocean gliders. Unpubl. MSc Thesis, Dalhousie University, Canada. Kämpf, J., and A. Kavi (2017), On the "hidden" phytoplankton blooms on Australia's southern shelves, Geophys. Res. Lett., 44, 1466–1473, doi: 10.1002/2016GL072096. Rossi, V., M. Feng, C. Pattiaratchi, M. Roughan, and A. M. Waite (2013), On the factors influencing the development of sporadic upwelling in the Leeuwin Current system, J. Geophys. Res. Oceans, 118, 3608–3621, doi:10.1002/jgrc.20242. Thomson, P.G., Mantovanelli, A., Wright, S.W., Pattiaratchi, C.B. (2015). In situ comparisons of glider bio-optical measurements to CTD water properties. Australian Marine Sciences Conference, Geelong, Victoria, July 5th – 9th 2015.

[Figure]

**Fig. 1.** Example of Figure with contours

---

## Author Comment (AC2) · 19 Jan 2019

Response to reviewers: os-2018-115 (Submitted to Ocean Science)

Seasonal and inter-annual variability of water column properties along the Rottnest continental shelf, south-west Australia by Miaoju Chen, Charitha Pattiaratchi, Anas Ghadouani and Christine Hanson.

We would like to thank and acknowledge both reviewers and the editor for their careful reading and constructive comments on the manuscript. There were no public comments. We believe that we have addressed the issues raised by reviewers and the

proposed changes to the manuscript are detailed in this document. We trust that the reviewers and the editor will find that the suggested changes will make the manuscript to be suitable for publication.

Reviewer #2

(1) Abstract Sentences 2 and 3 needs integration. They can be mix together integrating the information to report.

As suggested also by Referee#1 – the abstract has been revised and shortened. Sentences 2 and 3 have been integrated.

(2) Introduction Line 2. I would prefer to start the sentence like. "Among phytoplankton pigments, chlorophyll.„."There are other pigments in phytoplankton. I think that the is chlorophyll a that was used as indicator of phytoplankton biomass or you use the total amount of chlorophyll? Depending of your answer change the sentence accordingly.

Chlorophyll a was used as indicator of phytoplankton biomass. The ocean glider measures chlorophyll fluorescence. We have revised the text as follows:

Among the phytoplankton pigments, chlorophyll a (denoted as chlorophyll in the following description), is an important biological indicator of phytoplankton biomass in the water column.

(3) On paragraph 25 and 30, the sentence starting by "It is major is a mechanism. . ." I think this is a typing error remove the second "is" and "a".

We agree: the second "is" has been deleted.

(4) Page 5 paragraph 10. Please remove "highly" from the sentence Thus the RCA is a highly nutrient". . .

We agree: the word "highly" has been deleted.

(5) Page 5 paragraph 20. ". . .are not weather limited." The gliders were able to fly

even during bad weather conditions and strong winds? If so why you mentioned that in some seasons data were not present and the graphs have gaps?

It is true that gliders are capable of collecting data in harsh weather conditions and strong winds when ship sampling is not feasible, making them ideal platforms for sustained ocean surveys. However, due to funding limitations and operation reasons (i.e. gliders were not deployed over the Christmas/New Year holidays). The sampling was started with an ambitious goal of having at least one glider operating along the sampling line at all times. This was a quite a challenge and then with decreasing funding the deployments reduced to bi-monthly and then quarterly.

The gaps present in the graphs is due to the sampling nature of the gliders that travel in a saw-tooth pattern. In shallow water the down-cast and up-casts are close together whist in deeper waters the spacing is larger resulting in missing data.

(6) Methods Page 6. Paragraph 10. Of course, this methodology is a huge advance that regular measurements performed but I was also wondering if two or three days took to complete the transect was not too much time. In very dynamic areas, like upwelling areas, you might have complete different conditions within the 3 days for adjacent areas.

We agree with the review that in very dynamic areas, water properties may change within the 3 days, which we have submitted a paper to JGR (oceans) indicating that ocean gliders are capable of capturing diurnal upwelling.

However, in this study, we are addressing the seasonal and inter-annual variability, which will not be affected by these shorter term dynamic processes. In contrast, the ocean gliders provide high spatial resolution data (∼1-2 km intervals).

In addition, we have compared glider transect with satellite data (for both temperature and chlorophyll), and the surface features were consistent.

(7) Page 6 paraph 20. Did you performed any inter- calibration exercise between
the data collected from the chlorophyll fluorescence recorded and the quantification of chlorophyll (mg/m3). Again total chlorophyll or chlorophyll a only.

As part of the IMOS ocean glider program we have undertaken many inter-calibration exercises. Here, we attached a glider to a rosette sampler and collected concurrent data from the glider and niskin bottles at surface, mid-depth and bottom of the water column in 100m depth. The water samples were subjected to HPLC analyses to determine the Chlorophyll a concentrations. The comparison between ocean glider derived fluorescence and the HPLC Chlorophyll a concentrations was very good with r2 > 0.75 (n > 100) in the range 0.17 to 0.21 (mg m-3) (see Thomson et al., 2015).

A recent study by Beck (2016) found that, through inter-comparison of chlorophyll a and Wetlabs ECOPUCK derived fluorescence on ocean gliders, the original manufacturer's recommendation for the estimation of chlorophyll a from fluorescence provided the best estimate.

See also response to Reviewer#1 (point 5)

(8) Page 6. Last paragraph is very confusing. I don't understand if you reach the conclusion that the data was anormal by subtracting to the seasonal mean. You say previously that you perform quality control on data. Why you don't exclude the anormal data there. . ..so you won't have to deal with them again later. I think that you should try to rephrase and clarify that paragraph.

We believe that the reviewer meant page 7 (and not page 6) about the calculation of the anomalies. We do not discuss 'anormal' but 'anomalies' (def: something that deviates from the mean). We decided to present the data as anomalies as the seasonal variation in properties obscured the variability. A detailed explanation is also included as the response to Reviewer#1 and we have included a Table of the mean values (see above). To avoid confusion and for completeness we have included the distribution of absolute values as supplementary material.

We have rephrased and re-written the paragraph – we believe that inclusion of Table 1 and the absolute values as supplementary material will remove this confusion.

(9) Results Page 8. 3.3 Methods section; first line. This information must be in Methods section and justified why there are no mean for the season for that months, if you claim that gathering of data using gliders are not weather limited.

We have moved the sentence to the methods section.

Glider data were not available mainly due to operational reasons (e.g. funding, holiday season etc) rather than limited by weather – see also response above (point 5).

(10) Page 9 Paragraph 25 Typing error. Replace 6a by 7a.

We agree: have replaced 6a by 7a.

Page 10. The 3.5 section of results were very difficult to follow because the figure 9 were not understandable and I advise rebuilding it in clearer manner. Probably because I was confused with the figure I think that the paragraph 20 description was not correct. You have a higher chlorophyll concentration than 0.81 mg/m3 mentioned for 2009, between 10 and 20 m depth with chlorophyll concentration ranging 1.8mg/m3.

We have added additional text to make the figure clearer. The figure shows the depth mean values of temperature, salinity and fluorescence with time for all of the ocean glider transects. Figure 9 shows the same information as Figure 10 which the Reviewer stated to be very good as good evidence as different patterns between El Niño e La Niña (point 11 below). The difference is that Figure 9 shows the variation across the whole continental shelf whilst Figure 10 is the same data but at a particular distance (10 km). We have explained the relationship between the two figures in the text. As such we propose to retain this figure in the revised manuscript.

(11) Figure 10 was very good evidencing the different pattern between El Niño e La Niña.

Thank you – and we of course agree with the reviewer. No change required needed.

(12) Discussion Page 12. Line 14 (I think) Typing error : repeated.

We agree: have revised the typing error as follows:

The chlorophyll variability was related to the changes in the temperature, salinity distribution, which was linked to changes in the physical forcing: (1) the local wind field; (2) the Leeuwin current system; and, (3) air–sea fluxes, especially in terms of surface cooling and evaporation.

(13) Paragraph 20 to 25 must be in introduction.

We agree: have moved to the Introduction

(14) Paragraph 35 to 40 was already described in introduction.

We have deleted the paragraph

(15) Paragraph 25 and 35 must integrated with the obtained data by proving examples of the physical processes I think that your discussion must be improved by comparing your data with another data from upwelling coastal areas also impacted by El Niño and La Niña and compared the impact results in terms of chlorophyll and consequently in productivity between areas or with former events. It is very important to bring awareness of climate change and the huge effects they have in coastal dynamics and phytoplankton biomass and overall productivity giving relevant to studies like the ones you developed.

Please also see the response to Reviewer#1 – Point 7.

One of the unique features of the study region is that it does not follow well established processes and seasonality in other regions globally. Although the study region is located in an eastern ocean basin – it is not a major upwelling region (similar say to off Peru/Chile or South Africa). This is mainly because of the presence of the Leeuwin Current which flows southwards against the prevailing upwelling favourable winds that

promotes downwelling. During the summer there is shallow upwelling that results in elevated chlorophyll but is not able to sustain a large fishery. Also, maximum chlorophyll presented here are factor 10 lower than those observed off South Africa. In addition to this the maximum chlorophyll levels are observed during late autumn/early winter and thus not during the period upwelling favorable southerly winds are present. In summer the most persistent feature was the sub-surface chlorophyll maximum. Autumn is characterised by low wind speeds and winter has not prevailing wind (see Figure 2 and 3). Although other upwelling regions such as off Peru and South Africa does respond to ENSO events- mainly due to changes in the wind field, here the response is mainly due to changes in the strength of the Leeuwin Current that determines changes in the chlorophyll rather than upwelling. We have included this in the discussion.

We agree with the reviewer that climate change and the huge effects they have in coastal dynamics and phytoplankton biomass are very important and we have also included an additional paragraph to highlight that understanding inter-annual variability provides a good indication of what we may expect from climate change.

References: Beck M. (2016), Defining a multi-parameter optics-based approach for estimating Chlorophyll a concentration using ocean gliders. Unpubl. MSc Thesis, Dalhousie University, Canada. Kämpf, J., and A. Kavi (2017), On the "hidden" phytoplankton blooms on Australia's southern shelves, Geophys. Res. Lett., 44, 1466–1473, doi: 10.1002/2016GL072096. Rossi, V., M. Feng, C. Pattiaratchi, M. Roughan, and A. M. Waite (2013), On the factors influencing the development of sporadic upwelling in the Leeuwin Current system, J. Geophys. Res. Oceans, 118, 3608–3621, doi:10.1002/jgrc.20242. Thomson, P.G., Mantovanelli, A., Wright, S.W., Pattiaratchi, C.B. (2015). In situ comparisons of glider bio-optical measurements to CTD water properties. Australian Marine Sciences Conference, Geelong, Victoria, July 5th – 9th 2015.

---

## Author Response (AR2)

**Point-by-point response to editor**

We appreciate your comments, which are all valuable and helpful for revising and improving our manuscript. We have considered your comments carefully and have made appropriate corrections. As you can see from the revised version of our manuscript, your very constructive comments have been incorporated into the revision.

In the following, black indicates the comments by the editor; blue is our response to the reviewer; and note that all changes are highlighted by red in the manuscript.

(1) Following is the revisions and response to a few comments that the reviewer pointed out:

(2) I think the structure of the Introduction should be improved. Paragraphs with info on the biological processes should be following each other and the same holds for those treating physical processes.
Thank you for your comments. The structure of introduction has been reorganized as follows with paragraphs relating to biological processes mentioned first and (now the first 3 paragraphs) followed by the physical processes.

(3) Please write units with negative exponents throughout the manuscript, e.g. m s$^{-1}$
We have read through the entire manuscript changed nits to negative exponents.

(4) Language use is sloppy. In many cases, part of which I indicate below, the sentences are not logical and carry syntax errors. Please go through the text.
Thank you for the comments. We have read through the entire manuscript, modified the parts you have indicated and made additional changes.

(5) P3, L5 1953 (not 1985)
Corrected.

(6) P3, L7-8 " Seasonal cycles of phytoplankton concentrations signify the annual growth activity in pelagic systems" Is this a correct sentence? In particular the word "signify" seems strange here.
The sentence has been corrected as follows:
Seasonal cycles of phytoplankton concentrations are identifiable signals of the annual growth activity in pelagic systems (Cebrián and Valiela, 1999; Winder and Cloern, 2010).

(7) P3, L9 "The most common cycle is the spring bloom …" This is incorrect. A spring bloom cannot be a cycle. Please change wording.
The sentence has been corrected as follows:
In many parts of the world a spring bloom—an increase in phytoplankton concentrations in response to seasonal changes in temperature and solar radiation—is common and is usually present for a few weeks to months…

(8) P3, L11 Often a secondary peak in production develops … (add "in production" because the reader wouldn't know what kind of peak is meant.
Added.

(9) P3, L39 insert "during" after Sv
Corrected.

(10) P4, L14, 19 m s-1 (format with negative exponent)
Corrected.

(11) P4, L16-17 "Local sea breezes, superimposed on synoptic southerly winds (with speeds often >15 m/s), are prevalent in austral summer and spring (Pattiaratchi et al., 1997)." Delete this sentence. Exactly this info was given in lines 13-14
The whole sentence has been deleted.

(12) P4, L17-18 Change to: Storm systems in winter are associated … (As it is in the manuscript, it is again repeating the info above)
Corrected

(13) P4, L19 delete "storm"
Deleted

(14) P4, L20-21 Storms cannot have winds. Please change, for example: In the study region a typical pattern in winter is strong north/north-easterly winds blowing for 12 to 52 hours,
The whole sentence has been changed:
In the study region winter storms have a typical pattern with strong north/north-easterly components blowing for 12 to 52 hours,

(15) P4, L22-23 Dito: "Summer storms have southerly winds over a period of 3-4 days …" Storms cannot have winds. Change to: Summer storms are southerly over a period of 3-4-days …
The whole sentence has been changed:
Summer storms are southerly over a period of 3-4- days …

(16) P4, L24-25 This is a strange sentence. Above it was written that winds are high in winter, and here that there are calm conditions. Please change, for example: Calm conditions (<5 m s-1) are observed between winter storm fronts during autumn and winter (March-August).
Thank you for your comments. The whole sentence has been changed as follows:
Calm conditions (<5 m s$^{-1}$) are observed between winter storm fronts during autumn and winter (March-August).

(17) P5, L34-36 "Understanding … shelf." This is repeated info from lines 30-32

The whole sentence has been deleted.

(18)    P6, L20-21 delete "Teledyne Webb Research Slocum Electric" (repeated info from above)
        Deleted

(19)    P6, L38 Please define QA/QC
        QA/QC has been defined as Quality Assurance and Quality Control (QA/QC)

(20)    P7, L4 please use ""°C" not °Celsius
        Corrected.
(21)    P7, L11 "errors" instead of "faults"
        Corrected.

(22)    P7, L14 delete "after"
        Deleted.

(23)    P7, L24 delete "ensure ongoing reliability"
        Deleted.

(24)    P8, L13-14 "Summer storms, which usually lasted 36 hours, caused strong, southerly
        winds (> 25 m/s)" Storms cannot cause winds. Please change wording.
        The whole sentence has been changed as follows:
        Summer storms, which were usually associated with southerly winds, lasted up to 36 hours
        with speeds > 25 ms$^{-1}$).

(25)    P8, L33 "at lower depths" It is not clear to me which depths are meant here. Do you mean
        greater depths further offshore? Please make clear.
        The whole sentence has been changed as follows:
        On the upper continental shelf (< 40 m depth), the inshore waters were cooler and less saline
        than the offshore waters (Figure 4a).

(26)    P8, L36 delete "layer"
        "layer" has been deleted.

(27)    P9, L3 delete "depth"
        "depth" has been deleted.

(28)    P9, L32 I think "monthly mean water masses" is not the correct expression. What you
        mean is Monthly mean temperature and salinity (which describe water masses)
(29)    Corrected.

(30)    P9, L33 delete "year round," If there are 4 from 12 exceptions, year round is not
        appropriate.

"year round," has been deleted.

(31)    P9, L35 "indicated" instead of "showed"?
"showed" has been changed to "indicated".

(32)    P9, L36 "were warmer" than what?
The sentence has been changed as follows:
the inshore waters (< 40m depth) increased (~21.1–23.0°C).

(33)    P9, §3.3 Please indicate for what regions and their boundaries and depth ranges the
monthly means were calculated.
Inshore and offshore region has been defined as follows:
The inshore waters were defined as region where depth < 40m, and offshore waters were
defined as region where depth >40m.

(34)    P10, L5 Please refer to Fig. 5
Figure number added as follows:
…revealed significant seasonal variability (Figure 5a).
….and lowest in February (minimum of 0.43 mgm$^{-3}$, Figure 5b).

(35)    P10, L12 and further. In Figs 6, 7 and 8 the anomalies are shown against the seasonal
average. The sections in the 4 seasons do not have the same spatial extent. How does that
influence the seasonal average and its anomalies?
It has no influence as the anomalies were based on the mean (see Table 1) and as such it does
not depend on the length of the sections.

(36)    P10, L19-20 "During spring, the temperature anomaly indicated that water to be vertically
well mixed on the continental shelf (Figure 6a) with warmer water offshore" I think this
conclusion is not allowed from this figure. The figure only says that the anomaly is not big on
the shelf. Theoretically it is possible to have a stratified water column on the shelf, which is
present during all seasons, and then the anomaly would also be small. Whether the water
column is well mixed should be concluded from single sections. Second, are you writing here
that the water on the continental shelf be mixed with warmer water offshore? The sentence is
not unambiguous.
We agree and have modified the sentence by also relating it to the absolute values now
presented as supplementary figures.  They indicate the structure better.
During spring, the temperature distribution indicated that water to be vertically well mixed on
the upper continental shelf (Figures 6a; S1a) and vertically stratified in depths >40 m. The
offshore water in the upper layer was warmer compared to those on the inner shelf.

(37)    P11, L7 "This SCM was associated with the temperature and salinity distribution" This is
not clear. Please write how it was associated
A subsurface chlorophyll maximum (SCM) at ~100 m depth was present in the offshore

waters aligned with the vertical stratification in temperature and salinity and density (Figures 6a,7a; S4b).

(38) P11, L16 concentration
changed "concentrations" to "concentration".

(39) P11, L19-20 delete "The data highlights the interannual variability." (repeated info)
The sentence has been deleted.

(40) P12, L22-23 "The water column was vertically stratified because of the presence of a DSWC on the upper shelf and a thermocline in the offshore waters …" This sentence says the water column was stratified because of a thermocline. This is circular reasoning. Please modify.
Sentence changed:
The water column was vertically stratified across the whole transect with a DSWC present on the upper shelf (Figures 12a,b). Higher chlorophyll water was present in the DSWC's bottom layer. In the deeper waters, the higher chlorophyll water was associated with the SCM (Figures 12m, n).

(41) P12, L27-28 The winds were initially southerly and then changed to westerly and continued to be that way to 23 April … (add "continued to be that way", because I think this is what you mean; if you do not add that, you just say that the winds continued to 23 April)
Thank you for your comments.
"…continued to be that way" has been added.

(42) P12, L28-30 "On the upper shelf, the DSWC was eroded such that by 25 April, the temperature and salinity were vertically mixed in the water column (Figures 12e, k)." Which temperature and salinity? You cannot mix temperature and salinity, only water can be mixed.
The whole sentence has been changed as follows:
By 25 April, the water column on upper shelf was vertically mixed (Figures 12e, k).

(43) P12, L30-31 "the salinity on the upper shelf was vertically stratified" Only the water column can be stratified.
The whole sentence has been changed as follows:
On 28 April, when the winds decreased, the water column on the upper shelf was vertically stratified in salinity (Figure 12l).

(44) P14 I think the first paragraph of the Discussion can be shortened significantly without losing information.
Thank you for your comments. We agree the Discussion can been shortened, and following sentences has been deleted.
We have deleted several sentences.

(45)    P14, L5 (relatively high costs, and …
"high" has been added.

(46)    P14, L6 seasonal
season has been corrected to seasonal.

(47)    P14, L8 insert "layers" after sub-surface
"Layers" has been added – "in the sub-surface layers is unknown".

(48)    P14, L11 Please avoid using "In this paper" again
"In this paper" has been changed to "Here".

(49)    P14, L15 insert "also" after have
"also" has been added.

(50)    P14, L31 "the river discharge is mainly deflected south in winter" Is there a reference for
this contention?
Reference has been added.

(51)    P14, L38 "whilst the salinity increased" From which value are time it increased?
The sentence has been changes as follows:
whilst the salinity increased from January to March

(52)    P14, L40 "due to heat loss to the atmosphere" instead of "due to atmospheric heat loss"?
"due to atmospheric heat loss" has been changed to "due to heat loss to the atmosphere"

(53)    P15, L35 … seasonality like in other regions …
The sentence has been changed as follows:
seasonality like in other regions globally

(54)    P16, L4 "Thus" instead of "This"
"This" has changed to "Thus".

(55)    P16, L5 insert comma after "this" for enhancing readability
Comma has been inserted.

(56)    P16, L15 "highlighted" not "lighted"
Corrected to "highlighted".

(57)    P17, L6-7 "Local and basin-scale ocean forcing affected the coastal hydrography
(temperature and salinity) and biological variables (chlorophyll)." This is repeated info and
can be deleted at this place.

The whole sentence has been deleted.

(58)    P17, L16-24 Info in this paragraph appears verbatim in the Introduction. Please modify and shorten and avoid duplication.
We agree. And the whole sentence has been changed:
This event increased the Leeuwin current's volume transport in February and caused high sea surface temperature anomalies (~5 °C higher than normal; Feng et al., 2013).

(59)    P17-18, L34-2 This paragraph contains much info, about biological aspects, that is not relevant for the manuscript. Please delete or shorten strongly.
We agree. Changes has been made as follows:
First sentence: The extended impacts of the heat wave have significant ecological impact, causing temperate reef communities in Western Australia (Wernberg et al., 2016).
"Here the marine heat waves forced the contraction of a …. and tropical waters." has been deleted.

(60)    P18, L4-5 delete "was used to study the seasonal and interannual variability across the Rottnest continental shelf and". The sentence as it is in the text is awkward.
"was used to study the seasonal and interannual variability across the Rottnest continental shelf and" has been deleted.

(61)    P18, L7 delete "(DSWC)" (this has been defined before)
"(DSWC)" has been deleted.

(62)    P18, L8 delete "(ENSO)" (this has been defined before)
"(ENSO)" has been deleted.

(63)    The Conclusions section is mainly a summary. It is clear that some summing up will likely be needed in the Conclusions section, but I think here it is too much. The sentence towards the end of the section "It is concluded that the observed seasonal and interannual variability in chlorophyll concentrations were related to the changes in physical forcing." Is the first concluding remark. However, it is trivial. Please restructure the whole section, so that both summary and real conclusions are in balance.
The conclusions section has been re-written with the conclusions separated into dot points:
A seven-year, high-resolution ocean glider dataset indicated that the temperature, salinity, and chlorophyll a distribution along the Rottnest continental shelf exhibited strong seasonal and interannual variability.  Based on the results of the data analysis the following conclusions were reached:

• The seasonal variability was controlled by changes in the physical forcing that included winds,

formation of dense shelf water cascades and strength of the Leeuwin Current. The chlorophyll concentrations were higher during autumn and winter across the whole transect.

*Inner continental shelf (< 50 m)*:

- During spring and summer months, the water column was vertically well mixed due to strong wind mixing. In autumn and winter, DSWCs were the main physical feature.
- Chlorophyll concentrations were higher closer to the sea bed than at the surface during spring, summer, and autumn. In winter the chlorophyll concentrations were uniform through the water column.

*Outer continental shelf (> 50 m):*

- During spring and summer months the water column was vertically stratified in temperature that contributed to the formation of a subsurface chlorophyll maximum (SCM).
- With the onset of storms in autumn, the water column was well mixed with the SCM absent.

• Inter-annual variation was associated with ENSO events. Lower temperatures, higher salinity, and higher chlorophyll concentrations were associated with the El Niño event in 2010. During the strong La Niña event in 2011, temperatures increased and salinity and chlorophyll concentrations decreased. Over subsequent years, the temperatures gradually decreased, the salinity increased, and the chlorophyll concentrations continued to decrease. In autumn of 2014, the chlorophyll concentrations increased.

(64)     Table 1 °C, not °Celsius
°Celsius has been changed to °C

(65)     Figure 2 I am not sure everything in the wind roses will be readable. Please could you increase the fonts?
The font size has been increased.

(66)     References
Below there are some examples of incomplete references. Please go through the reference list for other possible errors.

Feng, 2003 should be:
Feng, M., G. Meyers, A. Pearce, and S. Wijffels (2003), Annual and interannual variations of the Leeuwin Current at 32°S, J. Geophys. Res., 108, 3355, doi:10.1029/2002JC001763, C11.

Field et al pages 237-240

Kilpatrick et al: Journal of Geophysical Research: Oceans, 123, 7550–7563. https://doi.org/10.1029/2018JC014248

Pattiaratchi, C. and Buchan, S: Reference is incomplete

We have gone through the whole references, and made several corrections in addition to your comments.